# PUSHING THE LIMIT OF SAMPLE-EFFICIENT OFFLINE REINFORCEMENT LEARNING

## ABSTRACT

Offline reinforcement learning (RL) has achieved notable progress in recent years. It enables learning optimized policy from fixed offline datasets and, therefore is particularly suitable for decision-making tasks that lack reliable simulators or have environment interaction restrictions. However, existing offline RL methods typically need a large amount of training data to achieve reasonable performance, and offer limited generalizability in out-of-distribution (OOD) regions due to conservative data-related regularizations. This seriously hinders the usability of offline RL in solving many real-world applications, where the available data are often limited. In this study, we introduce a highly sample-efficient offline RL algorithm that learns optimized policy by enabling state-stitching in a compact latent space regulated by the fundamental symmetry in dynamical systems. Specifically, we introduce a time-reversal symmetry (T-symmetry) enforced inverse dynamics model (TS-IDM) to derive well-regulated latent state representations that greatly ease the difficulty of OOD generalization. Within the learned latent space, we can learn a guide-policy to output the latent next state that maximizes the reward, bypassing the conservative action-level behavior constraints as used in typical offline RL algorithms. The final optimized action can then be easily extracted by using the guide-policy's output as the goal state in the learned TS-IDM. We call our method Offline RL via **T**-symmetry **E**nforced **L**atent **S**tate-Stitching (TELS). Our approach achieves amazing sample efficiency and OOD generalizability, significantly outperforming existing offline RL methods in a wide range of challenging small-sample tasks, even using as few as 1% of the original data in D4RL tasks.

## 1 INTRODUCTION

Offline reinforcement learning (RL) has seen rapid progress in recent years. It bypasses the reliance on environment interactions as in online RL methods, directly utilizing pre-collected offline data for policy learning, thus being ideal for many real-world tasks that lack high-fidelity simulators or have environment interaction restrictions (Levine et al., 2020; Fujimoto et al., 2018; Zhan et al., 2022). However, offline RL is also known to be prone to value overestimation, caused by extrapolation error when evaluating out-of-distribution (OOD) samples and amplified through the interactive dynamic programming procedure in RL (Kumar et al., 2019; Fujimoto et al., 2019). In the past few years, quite a few offline RL methods have been proposed, which commonly adopt the pessimism principle using strategies such as adding explicit or implicit policy constraints to prevent the selection of OOD actions (Kumar et al., 2019; Fujimoto et al., 2019; Wu et al., 2019; Fujimoto and Gu, 2021), penalizing value function on unseen samples (Kumar et al., 2020; Xu et al., 2022b; Bai et al., 2021; Lyu et al., 2022), or adopting in-sample learning to implicit regularize policy optimization (Kostrikov et al., 2021b; Xu et al.; Mao et al., 2024). What's in common with these methods is the use of some kind of action-level constraints to avoid OOD exploitation. Although this could stabilize offline value and policy learning, it inevitably leads to over-conservatism and crippled OOD generalization performance (Li et al., 2022; Cheng et al., 2023). Most of the existing offline RL methods only perform well when trained in sufficiently large amounts of offline data with reasonable state-action space coverage (e.g., 1 million samples for simple D4RL tasks (Fu et al., 2020)). This forms a stark contrast to the reality of most real-world scenarios, where the historical data are often limited and scaling up data collection can be rather costly (Zhan et al., 2022; Cheng et al., 2023). Hence although

offline RL is initially proposed to solve a wide range of real-world tasks, we still haven't seen too many practical deployments to date.

Enhancing the sample efficiency and OOD generalization capability is essential to making offline RL widely applicable to real-world applications. This is particularly important for small dataset settings, as most of the state-action space will become OOD regions. Several recent attempts have been made to improve the generalization performance of offline RL, which mainly follows three directions. The first direction builds upon the empirical observation that deep value functions interpolate well but struggle to extrapolate, thus allowing exploitation on interpolated OOD actions to promote generalization (Li et al., 2022). However, this method has a smoothness assumption on the offline dataset geometry and only applies to continuous action space. The second class of methods avoids the conservative action-level constraint and instead performs reward maximization on the state-space (Xu et al., 2022a; Park et al., 2024), which allows exploitation of OOD actions as long as the corresponding state transitions are reachable (also referred to as "state-stitching" (Xu et al., 2022a)). Although these methods enable promising state generalization capability, they still require the state-action space to have reasonable dataset coverage to enable valid state-stitching. The last and also the most explored direction is to learn compact and robust latent representations that facilitate generalization (Laskin et al., 2020; Agarwal et al., 2021; Yang and Nachum, 2021; Uehara et al., 2021; Weissenbacher et al., 2022; Cheng et al., 2023). Most of the existing representation learning studies only focus on extracting statistical-level information from the data, using techniques such as contrastive learning (Laskin et al., 2020; Agarwal et al., 2021; Yang and Nachum, 2021; Uehara et al., 2021). Although useful for improving sample efficiency, these methods lack in-depth modeling of the underlying dynamics patterns inside the sequential data, thus struggling to provide generalizable information beyond data distribution. Some recent methods (Weissenbacher et al., 2022; Cheng et al., 2023) propose to extract fundamental symmetries of dynamics to facilitate policy learning, such as the time-reversal symmetry (T-symmetry) in Cheng et al. (2023) (i.e., the underlying physical laws should not change under the time-reversal transformation: $t \to -t$). If we can find and leverage such universally held symmetries in the dataset, then it is possible to maximally promote OOD generalization without being restrained by data distribution-related information. Although promising, these methods are built upon existing action-level constraint offline RL backbone algorithms like CQL (Kumar et al., 2020) or TD3+BC (Fujimoto and Gu, 2021), which still suffer from the over-conservatism issue.

In this paper, we find that enabling state-stitching in a coherent, fundamental symmetry-enforced latent space can actually lead to a surprisingly strong sample-efficient offline RL algorithm. We refer to our method as Offline RL via **T**-symmetry **E**nforced **L**atent **S**tate-Stitching (TELS). Specifically, we introduce a T-symmetry enforced inverse dynamics model (TS-IDM) to not only learn well-behaved T-symmetry consistency representations that greatly alleviate the difficulty of OOD generalization, but can also facilitate effective action inference. Within the learned latent state space, we can optimize a T-symmetry regularized guide-policy to output the next state that maximizes the reward, bypassing the conservative action-level behavioral constraints as used in typical offline RL algorithms. Lastly, the optimized action can be easily extracted by plugging the output of the guide-policy as the goal state in the learned TS-IDM. The resulting algorithm achieves amazing sample efficiency and OOD generalization capability, significantly outperforming existing offline RL algorithms in a wide range of challenging reduced-size D4RL benchmark datasets, even using as few as 1% of the original samples. Our method greatly pushes the performance limit of offline RL under low data regimes, offering a new opportunity to tackle many previously unsolvable real-world tasks.

## 2 PRELIMINARIES

**Offline reinforcement learning.** We consider the standard Markov decision process (MDP) setting (Sutton and Barto, 2018), which is represented as a tuple $\mathcal{M} = \{\mathcal{S}, \mathcal{A}, r, \mathcal{P}, \rho, \gamma\}$, and a dataset $\mathcal{D}$, which consists of trajectories $\tau = (s_0, a_0, s_1, a_1, ..., s_T)$. Here $\mathcal{S}$ and $\mathcal{A}$ denote the state and action spaces, $r(s, a)$ is a scalar reward function, $\mathcal{P}(s'|s, a)$ and $\rho$ denote the transition dynamics and initial state distribution respectively, and $\gamma \in (0, 1)$ is a discount factor. Our goal is to learn a policy $\pi(a|s)$ based on dataset $\mathcal{D}$ by maximizing the expected return in the MDP: $\mathbb{E}[\sum_{t=0}^{\infty} \gamma^t \cdot r(s_t, a_t)]$.

**Offline policy optimization in the state space.** Instead of adopting conservative action-level constraints for offline policy learning, the recent Policy-guided Offline RL (POR) (Xu et al., 2022a) method proposes an alternative scheme, which decomposes the conventional reward-maximizing

policy into a guide-policy and an execute policy. The guide-policy only works in the state space to find the optimal next state that maximizes the state-value function, and the execute-policy is learned as an inverse dynamics model (Xu et al., 2022a) or a goal-conditioned imitative policy (Park et al., 2024). Such methods only need to learn a state-only value function $V$ using the IQL-style expectile regression (Kostrikov et al., 2021b) or the sparse value learning objective as discussed in Xu et al. (2021). We present the former as follows:

$$V = \arg\min_{V} \mathbb{E}_{(s,r,s')\sim\mathcal{D}} \left[ L_2^{\tau} \left( r(s) + \gamma \bar{V}(s') - V(s) \right) \right] \tag{1}$$

where $\bar{V}$ denotes the target value network and $L_2^{\tau}(x) = |\tau - \mathbb{1}(x < 0)|x^2$ denotes the asymmetric expectile regression loss as in IQL (Kostrikov et al., 2021a). Based on the learned state-value function, we can learn a guide-policy $\pi_g(s'|s)$ to serve as a prophet by telling which state the agent should (high reward) and can (logical generalization) go to, without being constrained to state-action transitions seen in the dataset. This can be achieved by leverage advantage weighted regression (AWR) type of objective (Neumann and Peters, 2008; Peng et al., 2019) to maximize the value while implicitly constraining $\pi_g$ to $s \rightarrow s'$ transitions observed in the dataset (i.e., state-stitching):

$$\pi_g = \arg\max_{\pi_g} \mathbb{E}_{(s,r,s')\sim\mathcal{D}} \left[ \exp(\alpha \cdot A(s,s')) \log \pi_g(s' \mid s) \right] \tag{2}$$

where the advantage $A(s,s') = r + \gamma V(s') - V(s)$ serves as the behavior cloning weight, and $\alpha$ is the inverse temperature parameter to prioritize value maximization over state-wise imitation learning.

For the execute-policy $\pi_e$, POR employs a supervised learning framework and trains $\pi_e$ by maximizing the likelihood of the actions given the states and next states: $\max_{\pi_g} \mathbb{E}_{(s,a,s')\sim\mathcal{D}}[\log \pi_e(a \mid s, s')]$. During evaluation phase, given the current state $s$, we can sample the optimized next state $s'$ from $\pi_g(s'|s)$, and can get final action simply as $a = \arg\max_a [\pi_e(a \mid s, \pi_g(s'|s))]$.

**Time-reversal symmetry for generalizable offline RL.** To extract well-behaved representations in small-sample settings, TSRL (Cheng et al., 2023) proposes to leverage the fundamental T-symmetry discovered in classical and quantum mechanics (Lamb and Roberts, 1998; Huh et al., 2020) to enhance the generalization of offline policy learning. Specifically, if we model the system dynamics with measurements $\mathbf{x}$ as a set of non-linear first-order differential equations (ODEs) expressed as $\frac{d\mathbf{x}}{dt} = F(\mathbf{x})$, a dynamical system is said to exhibit time-reversal symmetry if there is an invertible transformation $\Gamma$ that reverses the direction of time: $d\Gamma(\mathbf{x})/dt = -F(\Gamma(\mathbf{x}))$. TSRL introduces an extended definition of T-symmetry for discrete-time MDP setting, by learning a pair of ODE forward $F(s,a) \rightarrow \dot{s}$ and reverse dynamics $G(s',a) \rightarrow -\dot{s}$, and require them to satisfy $F(s,a) = -G(s',a)$, where the time-derivative of state $\dot{s} = \frac{ds}{dt}$ is approximated as $s' - s$.

Based on this intuition, TSRL constructed a T-symmetry enforced dynamics model (TDM) with an encoder-decoder architecture, to learn a pair of T-symmetry consistent ODE latent forward and reverse dynamics for representation learning. Although TSRL achieves impressive performance under small-sample settings, it still has limitations. First, TSRL only uses the learned encoder $\phi(s,a) = (z_s, z_a)$ from TDM to derive the latent representations, without fully exploiting the rich dynamics-related information in the model for downstream policy learning. Second, it needs simultaneous access to both state and action to derive latent representations, making Q-function maximization the only option for policy optimization, which inevitably requires adding conservative action-level constraints (like the behavior cloning term in TD3+BC (Fujimoto and Gu, 2021)) to stabilize training. Moreover, involving action as an input for representation learning is also prone to capturing the biased behaviors in the data-generating policy (e.g., data generated from expert policy will have special action patterns), which could impede learning fundamental, distribution-agnostic dynamics patterns in data.

## 3 OFFLINE RL VIA T-SYMMETRY ENFORCED LATENT STATE-STITCHING

We now present our proposed method, TELS, which comprises a T-symmetry enforced inverse dynamics model (TS-IDM) integrated with an effective offline policy optimization procedure operated in the latent state space. Inspired by previous work TSRL (Cheng et al., 2023), we design the TS-IDM to extract the fundamental, T-symmetry preserving representations from the limited data, which not only facilitates OOD generalization for policy learning, but can also be seamlessly used as a execute-policy for optimal action extraction. An overall illustration of our proposed framework is presented in Figure 1.

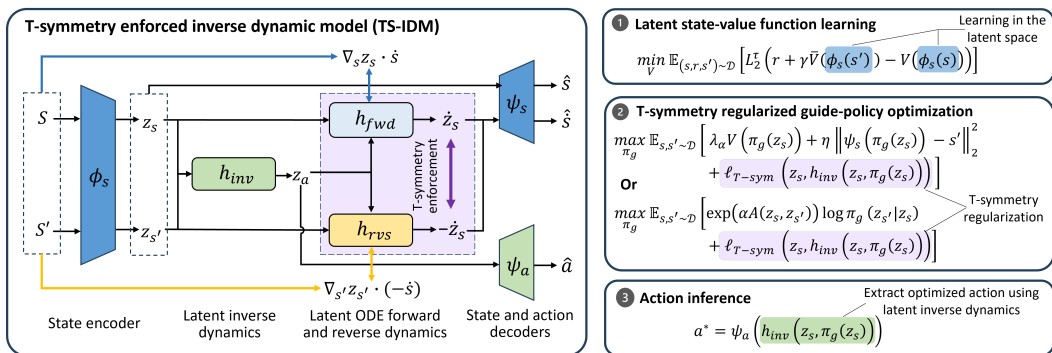

Figure 1: Overview of our proposed TELS framework.

### 3.1 T-symmetry Enforced Inverse Dynamic Model

If we look at the input and output of our proposed TS-IDM, it functions similarly to a typical inverse dynamics model that takes current and next state $(s, s')$ as input and outputs the action $a$. However, TS-IDM's architecture is distinct in several aspects. In its interior, it comprises a state encoder $\phi_s(s) = z_s$, a latent inverse dynamics model $h_{inv}(z_s, z_{s'}) = z_a$, a pair of T-symmetry enforced latent ODE forward and reverse dynamics models $h_{fwd}(z_s, z_a) = \dot{z}_s$ and $h_{rvs}(z_{s'}, z_a) = -\dot{z}_s$, an action decoder $\psi_a(z_a) = \hat{a}$, and an extra state decoder $\psi_s(z_s) = \hat{s}$. In the following, we describe their detailed design logic and learning objectives.

**State encoding and decoding.** As we have discussed before, constructing an informative and well-behaved latent space is crucial for effective offline policy optimization in the small-sample setting. Therefore, we introduce a state encoder $\phi_s(s) = z_s$ to embed the current and next states $(s, s')$ into latent space $(z_s, z_{s'})$, together with an extra state decoder $\psi_s(z_s) = s$ to map the latent representations back to the original state space, which ensures that the learned latent state representations do not become overly distorted. This implies the following state reconstruction loss:

$$\ell_{s\text{-}rec}(s) = \|\psi_s(\phi_s(s)) - s\|_2^2 \tag{3}$$

**Latent inverse dynamics model.** Inside the TS-IDM, we construct a latent inverse dynamics model $h_{inv}(z_s, z_{s'}) = z_a$, where $z_s$ and $z_{s'}$ serve as inputs to infer the latent action $z_a$. We employ an action decoder $\psi_a(z_a) = \hat{a}$ to map the latent actions back to the original action space, ensuring that the action representations are meaningful and interpretable. The loss term is as follows:

$$\ell_{inv}(s, a, s') = \|\psi_a(h_{inv}(z_s, z_{s'})) - a\|_2^2 \tag{4}$$

**Latent ODE forward and reverse dynamic models.** Drawing inspiration from previous research that integrates physics-informed insights into dynamical systems modeling (Mezić, 2005; Brunton et al., 2016; Champion et al., 2019; Cheng et al., 2023), we embed a pair of latent ODE forward and reverse dynamics $h_{fwd}(z_s, z_a) = \dot{z}_s$ and $h_{rvs}(z_{s'}, z_a) = -\dot{z}_s$ to separately capture the forward and reverse time evolution on states. We are interested in modeling ODE systems because it encourages learning parsimonious models helpful to uncover fundamental properties from the data (Brunton et al., 2016; Champion et al., 2019), that can maximally promote generalization. Note that based on the chain rule, we can derive the supervision signal for the latent dynamics models with $\dot{z}_s = \frac{dz}{dt} = \frac{dz_s}{ds} \cdot \frac{ds}{dt} = \nabla_s z_s \cdot \dot{s} = \nabla_s \phi_s(s) \cdot \dot{s}$ to enforce the ODE property. Therefore, we can use the following training losses for $h_{fwd}$ and $h_{rvs}$:

$$\ell_{fwd}(s, s') = \|(\nabla_s z_s)\dot{s} - \dot{z}_s\|_2^2 = \|\nabla_s \phi_s(s)\dot{s} - h_{fwd}(z_s, z_a)\|_2^2 \tag{5}$$

$$\ell_{rvs}(s, s') = \|(\nabla_{s'} z_{s'})(-\dot{s}) - (-\dot{z}_s)\|_2^2 = \|\nabla_{s'} \phi_s(s')(-\dot{s}) - h_{rvs}(z_{s'}, z_a)\|_2^2 \tag{6}$$

where the latent action $z_a$ is obtained from the latent inverse dynamics model $h_{inv}(z_s, z_{s'})$.

Note that in the above loss terms, we actually implicitly enforced the ODE property on the state encoder $\phi_s$, the same should also apply to the state decoder $\psi_s$ to ensure compatibility with the T-symmetry formalism, i.e. the time-derivative of the state encoder $\frac{d\phi_s(s)}{dt}$ and decoder $\frac{d\psi_s(z_s)}{dt}$

should behave in the same way as $\dot{z}_s$ and $\dot{s}$. Similar to the previous treatment on the state encoder, as $\dot{s} = \frac{d\psi_s(z_s)}{dt} = \frac{d\psi_s(z_s)}{dz_s} \cdot \frac{dz_s}{dt} = \nabla_{z_s}\psi(z_s) \cdot \dot{z}_s$, hence we can use the following objective to enforce the ODE property for the state decoder $\psi_s$:

$$\ell_{s\text{-}ode}(s, s') = \|\nabla_{z_s}\psi_s(z_s) \cdot h_{fwd}(z_s, z_a) - \dot{s}\|_2^2 + \|\nabla_{z_s}\psi_s(z_s) \cdot (-h_{rvs}(z_{s'}, z_a)) - \dot{s}\|_2^2 \quad (7)$$

In the above learning objective, we use the output $\dot{z}_s$ obtained from $h_{fwd}$ and $-h_{rvs}$ in our calculation to further regularize the output of the ODE forward and reverse dynamics. Again, the latent action $z_a$ is obtained from the $h_{inv}(z_s, z_{s'})$. With the above coupled design among the state encoder, state decoder, and the latent forward and reverse dynamics, we obtain a strongly consistent ODE system to capture the fundamental dynamics properties in the offline dataset.

**T-symmetry enforcement.** To further regularize the learned latent representations, we incorporate the extended version of T-symmetry (Cheng et al., 2023) by requiring $h_{fwd}(z_s, z_a) = -h_{rvs}(z_{s'}, z_a)$. We adopt a similar T-symmetry consistency loss as in TSRL (Cheng et al., 2023):

$$\ell_{T\text{-}sym}(z_s, z_a) = \|h_{fwd}(z_s, z_a) + h_{rvs}(z_s + h_{fwd}(z_s, z_a), z_a)\|_2^2 \quad (8)$$

where we use the fact that $z_{s'} = z_s + \dot{z}_s = z_s + h_{fwd}(z_s, z_a)$ and $h_{rvs}(z_s + h_{fwd}(z_s, z_a), z_a) = -\dot{z}_s = -h_{fwd}(z_s, z_a)$ to further couple the learning process of $h_{fwd}$ and $h_{rvs}$. Moreover, given a latent state-action pair $(z_s, z_a)$, the above T-symmetry consistency loss can also serve as an evaluation metric to assess their agreement with the learned TS-IDM. A large T-symmetry loss indicates that the latent state-action representation $(z_s, z_a)$ induced by some $(s, s')$ may not satisfy the fundamental dynamics pattern, therefore more likely to be a problematic or non-generalizable sample.

**Overall learning objective.** Finally, the complete training loss function of TS-IDM is as follows:

$$\mathcal{L}_{TS\text{-}IDM} = \sum_{(s,a,s') \in \mathcal{D}} \left[ \ell_{s\text{-}rec} + \ell_{inv} + \beta \cdot (\ell_{fwd} + \ell_{rvs} + \ell_{s\text{-}ode} + \ell_{T\text{-}sym}) \right] \quad (9)$$

Where $\beta$ is a hyperparameter that balance between capturing fundamental dynamic properties and ensuring interpretability of the learned representation. By enforcing both the ODE property and T-symmetry consistency in the inverse dynamics modeling, not only produces well-regularized and informative latent state representations but also reliable action prediction. In the next section, we will describe how different components in the learned TS-IDM can be seamlessly integrated to form an extremely sample-efficient offline policy optimization framework.

## 3.2 Sample-Efficient Offline Policy Optimization Within Latent Space

Once we have a learned TS-IDM, we can extract three highly useful components from it to facilitate sample-efficient downstream offline policy optimization, including 1) a robust state encoder $\phi(s)$ that provides a compact and generalizable latent space ideal for state-stitching; 2) T-symmetry consistency as an additional regularizer to prevent erroneous generalization when learning a guide-policy in the latent state space; and 3) the TS-IDM itself can serve as an execute-policy as in POR (Xu et al., 2022a) to extract optimized action given the learned guide-policy.

**Latent state-value functions learning.** Based on the state encoder $\phi_s(s)$ from the pre-trained TS-IDM, we can convert the entire offline policy optimization process into the latent state space, which enjoys both the stable learning process and generalizability due to more compact and well-behaved representations. Specifically, we can use a similar IQL-style expectile regression loss to learn a state-value function $V(z_s)$ as in Eq. (1), but in the latent state space:

$$V = \arg\min_V \mathbb{E}_{(s,r,s') \sim \mathcal{D}} \left[ L_2^\tau(r + \gamma \bar{V}(\phi_s(s')) - V(\phi_s(s))) \right] \quad (10)$$

**T-symmetry regularized guide-policy optimization.** A major benefit of learning within the T-symmetry preserving latent space is that, as T-symmetry captures what is essential and invariant about the dynamical system, thus it can generalize and provide information even for OOD samples beyond the offline dataset. This naturally favors learning a reward-maximizing guide-policy $\pi_g$ in the latent space, which can enjoy more effective state-stitching. Moreover, by leveraging the T-symmetry consistency term in Eq. (8) as an additional regularizer, we can prevent $\pi_g$ from outputting problematic and non-generalizable latent next state, thereby further enhancing logical OOD generalization.

---

**Algorithm 1** Offline RL via T-symmetry Enforced Latent State-Stitching (TELS).

---

**Require:** Offline dataset $\mathcal{D}$.
1: *// TS-IDM learning*
2: Learning the state encoder $\phi_s$, state decoder $\psi_s$, action decoder $\psi_a$, latent inverse dynamics $h_{inv}$, latent forward and reverse dynamics $h_{fwd}$ and $h_{rvs}$ using the TS-IDM learning objective Eq. (9).
3: Initialize $V_\theta, V_{\theta'}, \pi_\sigma$
4: *// Policy training*
5: **for** $t = 1, \cdots, M$ training steps **do**
6:     Sample transitions $(s, r, s') \sim \mathcal{D}$ and compute their representations $(z_s, z_{s'})$ using the state encoder $\phi_s$.
7:     Use $(z_s, r, z_{s'})$ to update the latent state-value function $V$ using Eq.(10)
8:     Use $(z_s, z_{s'})$ to update the latent guide-policy $\pi_g$ using Eq.(11) or (12).
9: **end for**
10: *// Evaluation*
11: Get initial state $s$ from environment
12: **while** not done **do**
13:     Get optimized next state $z_{s'}^*$ using guide-policy $\pi_g$.
14:     Extract action $a$ using Eq. (13)
15: **end while**

---

In this work, we provide two instantiations for deterministic and stochastic latent guide-policy learning, respectively. For deterministic guide-policy $\pi_g(z_s) \to z_{s'}^*$, we adopt the following policy optimization objective:

$$\pi_g = \arg\max_{\pi_g} \mathbb{E}_{(s,s')\sim\mathcal{D}} \Big[ \lambda_\alpha V(\pi_g(z_s)) + \eta\|\psi_s(\pi_g(z_s)) - s'\|_2^2 + \ell_{T\text{-}sym}(z_s, h_{ivs}(z_s, \pi_g(z_s))) \Big]$$ (11)

where $z_s = \phi_s(s)$. The above can be perceived as a TD3+BC style policy optimization objective (Fujimoto and Gu, 2021), but completely conducted in the latent state space. The first term maximizes the latent state-value function $V$, weighted by a normalization term $\lambda_\alpha = \alpha/[\sum_{s_i}|V(\phi_s(s_i))|/N]$ computed based on hyperparameter $\alpha$ and $N$ samples in the training batch. The second term regularizes the next state decoded from the guide-policy using state decoder $\psi_s$ should not deviate too much from the ground truth next state $s'$ in the dataset. The last term regularizes guide-policy induced latent state-action pair (i.e., $(z_s, z_a) = (z_s, h_{inv}(z_s, \pi_g(z_s)))$) to comply with the T-symmetry consistency specified in the learned TS-IDM.

For stochastic guide-policy $\pi_g(z_{s'}|z_s)$, we borrow the similar AWR-style policy optimization objective as in Eq. (2), but incorporating the T-symmetry consistent regularization:

$$\pi_g = \arg\max_{\pi_g} \mathbb{E}_{(s,s')\sim\mathcal{D}} \Big[ \exp(\alpha \cdot A(z_s, z_{s'})) \log \pi_g(z_{s'} \mid z_s) + \ell_{T\text{-}sym}(z_s, h_{ivs}(z_s, \pi_g(\cdot|z_s))) \Big]$$ (12)

where $z_s = \phi_s(s)$, $z_{s'} = \phi_s(s')$, and $A(z_s, z_{s'}) = r + \gamma V(z_{s'}) - V(z_s)$.

In our experiments, we find that the deterministic version objective Eq. (11) works well for the MuJoCo locomotion tasks, while the stochastic version Eq. (12) works better for more complex D4RL Antmaze tasks (Fu et al., 2020), potentially due to more stochastic nature of the task environment.

**Action inference.** After learning the guide-policy $\pi_g$, we can further use it to generate the optimized action for control. To do this, we can simply use the latent next state $z_{s'}^*$ outputted from the $\pi_g(z_s)$ or $\pi_g(\cdot|z_s)$ as the goal state, and plug it into the learned latent inverse dynamics model $h_{inv}(z_s, z_{s'})$ in TS-IDM to replace $z_{s'}$. The final action can be extracted by decoding the resulting latent action from $h_{inv}$ using the state decoder $\psi_a$ :

$$a^* = \psi_a(h_{inv}(z_s, \pi_g(z_s)))$$ (13)

Note that there is no training process needed for this stage. We fully utilize the learned TS-IDM to serve our purpose. We summarize the complete training and inference procedure of our framework in Algorithm 1.

## 4 EXPERIMENTS

In this section, we present the empirical evaluation results of TELS on the D4RL MuJoCo-v2 and Antmaze-v1 datasets (Fu et al., 2020) against with behavior cloning (BC), and SOTA offline RL

Table 1: Average normalized score on D4RL MuJoCo and Antmaze tasks in reduced-size datasets. We report some of the small-sample evaluation results from the TSRL paper (Cheng et al., 2023). The evaluation results under the full dataset are listed in Appendix B.

| Task | Size (ratio) | BC | TD3+BC | CQL | IQL | DOGE | POR | TSRL | TELS |
|------|-------------|-----|--------|-----|-----|------|-----|------|------|
| Hopper-m | 10k (1%) | 29.7±11.7 | 40.1±18.6 | 43.1±24.6 | 46.7±6.5 | 44.2 ± 10.2 | 46.4 ± 1.7 | 62.0±3.7 | **77.3 ± 10.7** |
| Hopper-mr | 10k (2.5%) | 12.1±5.3 | 7.3±6.1 | 2.3±1.9 | 13.4±3.1 | 17.9 ± 4.5 | 17.4 ± 6.2 | 21.8±8.2 | **43.2 ± 3.5** |
| Hopper-me | 10k (0.5%) | 27.8±10.7 | 17.8±7.9 | 29.9±4.5 | 34.3±8.7 | 50.5 ± 25.2 | 37.9 ± 6.1 | 50.9±8.6 | **100.9 ± 6.8** |
| Halfcheetah-m | 10k (1%) | 26.4±7.3 | 16.4±10.2 | 35.8±3.8 | 29.9±0.12 | 36.2 ± 3.4 | 33.3±3.2 | 38.4±3.1 | **40.8 ± 0.6** |
| Halfcheetah-mr | 10k (5%) | 14.3±7.8 | 17.9±9.5 | 8.1±9.4 | 22.7±6.4 | 23.4 ± 3.6 | 27.5±3.6 | 28.1±3.5 | **33.2 ± 1.0** |
| Halfcheetah-me | 10k (0.5%) | 19.1±9.4 | 15.4±10.7 | 26.5±10.8 | 10.5±8.8 | 26.7 ± 6.6 | 34.7±2.6 | 39.9±21.1 | **40.7 ± 1.2** |
| Walker2d-m | 10k (1%) | 15.8±14.1 | 7.4±13.1 | 18.8±18.8 | 22.5±3.8 | 45.1 ± 10.2 | 22.2±3.6 | 49.7±10.6 | **62.4 ± 5.3** |
| Walker2d-mr | 10k (3.3%) | 1.4±1.9 | 5.7±5.8 | 8.5±2.19 | 10.7±11.9 | 13.5 ± 8.4 | 14.8±4.2 | 26.0±11.3 | **54.8 ± 6.0** |
| Walker2d-me | 10k (0.5%) | 21.7±8.2 | 7.9±9.1 | 19.1±14.4 | 26.5±8.6 | 35.3 ± 11.6 | 20.1±8.6 | 46.4±17.4 | **87.4 ± 13.3** |
| Antmaze-u | 10k (1%) | 44.7 ± 42.1 | 0.7 ± 1.2 | 0.1 ± 0.0 | 65.1 ± 19.4 | 56.3 ± 24.4 | 6.1 ± 7.3 | 76.1 ± 15.6 | **88.7 ± 7.7** |
| Antmaze-u-d | 10k (1%) | 24.1 ± 22.2 | 16.27 ± 16.4 | 0.5 ± 0.1 | 34.6 ± 18.5 | 41.7 ± 18.9 | 42.1 ± 14.2 | 52.2 ± 22.1 | **60.9 ± 16.9** |
| Antmaze-m-d | 0.1M (10%) | 0.0 | 0.0 | 0.0 | 4.8 ± 5.9 | 0.0 | 0.0 | 0.0 | **47.2 ± 17.3** |
| Antmaze-m-p | 0.1M (10%) | 0.0 | 0.0 | 0.0 | 12.5 ± 5.4 | 0.0 | 0.0 | 0.0 | **62.9 ± 17.8** |
| Antmaze-l-d | 0.1M (10%) | 0.0 | 0.0 | 0.0 | 3.6 ± 4.1 | 0.0 | 0.0 | 0.0 | **39.8 ± 14.1** |
| Antmaze-l-p | 0.1M (10%) | 0.0 | 0.0 | 0.0 | 3.5 ± 4.1 | 0.0 | 0.0 | 0.0 | **47.3 ± 13.1** |

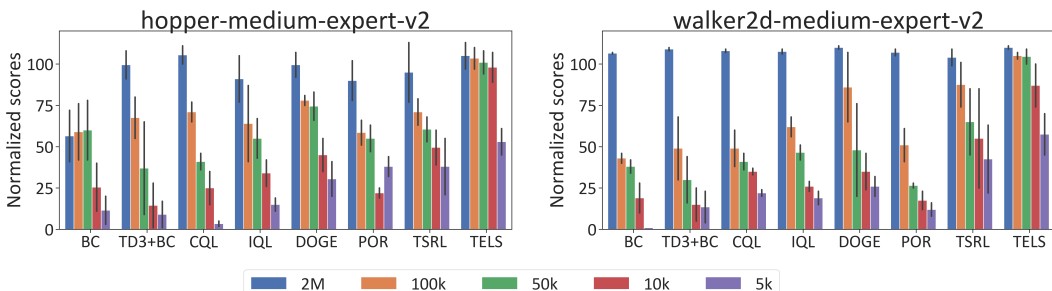

Figure 2: Performance of TELS against baselines under different data sizes.

methods TD3+BC (Fujimoto and Gu, 2021), CQL (Kumar et al., 2020), IQL (Kostrikov et al., 2021a), DOGE (Li et al., 2022), POR (Xu et al., 2022a), and TSRL (Cheng et al., 2023), which is the current SOTA method in small-sample settings. We then conduct several ablation studies to evaluate the functionality of each component of TS-IDM in section 4.2. Additionally, we demonstrate that the learned representation model significantly improves the performance of other baselines in small-sample settings. The implementation details, experimental setup, and additional results can be found in Appendix A and B.

## 4.1 PERFORMANCE COMPARISON ON SMALL-SAMPLE SETTING

We evaluate the performance of TELS against baseline methods on reduced-sized D4RL datasets ranging from 5k to 0.1M samples. We employ a similar approach utilized in TSRL (Cheng et al., 2023), by randomly sub-sample trajectories from the full datasets. The evaluation results reveal that most baseline methods fail to derive reliable policies under extremely small datasets due to insufficient samples to support action constraints or to effectively learn the value function. By leveraging the advantages of a T-symmetry enforced dynamics model, TELS extracts fundamental symmetries of the dynamics to facilitate policy learning, demonstrating effective performance even with limited data. However, as previously discussed, TSRL incorporates actions into its representation inputs, making it susceptible to capturing biased behaviors. Furthermore, it employs an action-level constraint structure, resulting in over-conservatism issues. These drawbacks become more pronounced, especially with low-quality datasets like the "medium-replay" dataset, whose data is mostly collected by random and low-performing behavior policies.

In contrast, TELS significantly outperforms all baselines in both MuJoCo locomotion and AntMaze tasks, demonstrating strong generalizability and superior sample efficiency under small-sample settings. It is notable that it can still retrieve performative policies from the "medium-replay" dataset.

This suggests that TS-IDM is able to extract reliable fundamental features from states without necessarily requiring high-quality data. Moreover, as data quality improves, TELS's performance dramatically surpasses that of the baselines, particularly in the "hopper-medium-expert" dataset, which reveals that the learned TS-IDM can provide a stronger representative latent space as the data quality increases. By incorporating a generative guide policy to sample valuable states for the inverse dynamics model, all these components together result in an extremely sample-efficient RL algorithm. We also conduct experiments on various data scales using the Walker2d-medium and Walker2d-medium-expert datasets separately to examine the impact of training data size on the algorithm's performance. From the results in Figure 2, we observe that TELS exhibits more stable performance and, even with only 5k samples, still surpasses other baseline approaches.

To further verify the generalizability of TELS, we conducted experiments on the AntMaze tasks within a small-sample setting, which presents additional challenges for learning reliable policies. Specifically, for the larger maps in the AntMaze-medium and AntMaze-large scenarios, the limited sampled datasets could contain a higher proportion of unsuccessful trajectories, thus requiring the policy to generalize to OOD regions to reach the goal state. For the small-sample conduct for the navigation task, as providing only 10k samples will lead to an overly sparse state space, we choose to sample about 0.1M dataset and evaluate the performance of the policy. It can be seen as an extremely difficult setting in which all baselines, including TSRL, fail to derive effective policies. In contrast, TELS is still able to produce well-performing policies. Additionally, we conducted a more challenging scenario by gradually removing portions of critical data points from the AntMaze-medium-diverse dataset, and We found that even after removing 90% of the data, TELS was still able to achieve the goal state, demonstrating its remarkable generalizability. We present more details in Appendix B.3.

## 4.2 INVESTIGATION ON TS-IDM

In this section, we evaluate the functionality of the main components of TS-IDM to determine which contributes most significantly to deriving an effective policy. We start with a basic autoencoder framework and incrementally add components until we develop TS-IDM. Throughout this process, we assess the performance at each stage to gain a clear understanding of how each component contributes to forming a well-behaved representation.

We perform the ablation studies on the "medium-expert" dataset using 10k data samples across three tasks. Here, we also demonstrate the procedure for constructing the TS-IDM from a basic autoencoder (AE) framework. We begin by encoding the states into a latent space using the autoencoder and generating actions through an inverse dynamics model, denoted as "AE $+h_{inv}$." We then learn the value functions and guide policy within the latent space to evaluate its performance. Next, we incorporate the forward and reverse dynamic models, $h_{\text{fwd}}$ and $h_{\text{rvs}}$, into the representation learning to capture more information about the dynamical systems. In this step, we employ conventional methods to train these dynamic models instead of using $\ell_{s\text{-}ode}$ for modeling the ODE systems. Subsequently, we enforce the ODE properties on the forward and reverse dynamics using $\ell_{s\text{-}ode}$ to capture the fundamental dynamical properties of the system. Finally, we involve $\ell_{T\text{-}sym}$ to couple the learning processes of $h_{\text{fwd}}$ and $h_{\text{rvs}}$, thereby enforcing the T-symmetry property on the latent representations. We present the implementation details in Appendix A.

Table 2: Ablations on the components of TS-IDM.

|  | Hopper-m-e | Halfcheetah-m-e | Walker2d-m-e |
|---|---|---|---|
| AE $+ h_{inv}$ | $17.2 \pm 7.0$ | $29.7 \pm 3.6$ | $24.5 \pm 10.1$ |
| $+ h_{fwd}, h_{rvs}$ | $35.5 \pm 7.3$ | $31.3 \pm 1.1$ | $33.6 \pm 9.2$ |
| $+ \ell_{s\text{-}ode}$ | $48.7 \pm 5.0$ | $34.2 \pm 1.2$ | $68.5 \pm 9.1$ |
| $+ \ell_{T\text{-}sym}$ | $100.9 \pm 6.8$ | $40.7 \pm 1.2$ | $87.4 \pm 13.3$ |

The results indicate that with limited datasets, a basic autoencoder framework fails to provide informative representations for downstream policy learning. Incorporating the dynamics model introduces some dynamic information about the system but remains insufficient to capture these features effectively within the small available dataset. As expected, by modeling with the ODE properties, the forward and reverse dynamics models can extract fundamental patterns from the limited data, enhancing the reliability of the learned representations. This improvement is particularly significant

in the "Hopper-medium-expert" and "Walker2d-medium-expert" datasets, where performance improves substantially. This demonstrates that learning system dynamics through ODEs yields more informative representations. Lastly, enforcing T-symmetry consistency results in the most informative representation space and provides a reliable regularizer for the downstream RL method to derive a performative policy.

### 4.3 INTEGRATING WITH OTHER BASELINE METHODS

In order to verify the usability of the learned TS-IDM, we utilize the learned TS-IDM to construct latent states for the two baseline methods, IQL (Kostrikov et al., 2021a) and DOGE (Li et al., 2022). Specifically, we train both baselines incorporating the pre-trained state encoder of TS-IDM, $\phi_s(s)$, to encode the original states into their corresponding latent representations.

Table 3: The performance of IQL and DOGE with or without the representation from TS-IDM.

|  | Hopper-medium | | Walker2d-medium | |
| --- | --- | --- | --- | --- |
|  | IQL | DOGE | IQL | DOGE |
| w/o TS-IDM repre | $46.7 \pm 6.5$ | $44.2 \pm 10.2$ | $22.5 \pm 3.8$ | $45.1 \pm 10.2$ |
| TS-IDM repre | $\mathbf{54.1 \pm 7.8}$ (↑16%) | $\mathbf{57.3 \pm 5.8}$(↑30%) | $\mathbf{41.3 \pm 9.3}$ (↑83%) | $\mathbf{48.7 \pm 14.5}$ (↑8%) |

The results in Table 3 demonstrate notable performance improvements when IQL and DOGE are trained using the latent representations derived from TS-IDM. In both the Hopper-medium and Walker2d-medium datasets, the applications of TS-IDM yielded significant performance improvement compared to the algorithms trained without this representation. For the Hopper-medium task, IQL and DOGE achieved relative performance improvements of 16% and 30%, respectively. Similarly, in the Walker2d-medium task, we observe even more pronounced improvements, with IQL showing an 83% increase in performance and DOGE demonstrating an 8% enhancement.

These results underscore the utility of the TS-IDM in facilitating more efficient learning by constructing a latent space that is both compact and structured. The learned representations not only improve the generalization capability of the policy but also mitigate challenges related to distributional shift problems, especially under limited data coverage. The consistency of performance improvements across both IQL and DOGE, despite their differing approaches to offline learning, further implies that TS-IDM is capable of providing generalizable and informative representation space under small-sample settings.

## 5 RELATED WORK

To mitigate the issue of the distributional shift in offline RL, several existing methods adopt the principle of pessimism by imposing explicit behavior regularization to explicitly penalize the action divergence (Wu et al., 2019; Kumar et al., 2019; Xu et al., 2021; Fujimoto and Gu, 2021), or implicit constraints the policy that prevents the agent from selecting OOD actions (Kumar et al., 2020; Xu et al., 2022b; Bai et al., 2021; Lyu et al., 2022). Alternatively, employ an in-sample learning approach to implicitly regularize policy optimization, which updates the critic functions without querying the generated actions to stabilize the learning process of the value function (Brandfonbrener et al., 2021; Kostrikov et al., 2021b; Xu et al.; Mao et al., 2024). There are also works incorporating the pessimistic constraints on the policy based on the uncertainty measurement (Wu et al., 2021; An et al., 2021; Bai et al., 2021).

Most methods tend to perform well when provided with a sufficiently large dataset and under the assumption of adequate state-action space coverage, which is often crucial for analyzing theoretical performance guarantees (Kumar et al., 2019; Chen and Jiang, 2019). However, in real-world scenarios, particularly when training data is limited, encountering a significant number of OOD regions is inevitable. Applying strict data-dependent regularizations in such cases can lead to substantial performance degradation and poor generalization.

In order to improve the generalizability of the policy under the OOD regions. One approach leverages the geometry with deep function approximators over the dataset, enables exploitation in generalizable OOD areas. Another strategy moves away from conservative action-level constraints, focusing on reward maximization within the state space. This "state-stitching" approach permits the exploitation

of OOD actions as long as their corresponding state transitions are feasible (Xu et al., 2022a; Park et al., 2024). While these methods demonstrate their generalizability over OOD states, they still require sufficient dataset coverage across the state-action space to support deriving a well-behaved policy. Additionally, much research has focused on learning compact latent representations to improve generalization (Laskin et al., 2020; Agarwal et al., 2021; Yang and Nachum, 2021; Uehara et al., 2021). Whereas these approaches demonstrate their ability to improve the sample efficiency, these methods lack in-depth modeling of the underlying dynamics patterns inside the sequential data, thus struggling to provide generalizable representations for the OOD data. In order to retrieve the more fundamental features of the dynamics, recent methods (Weissenbacher et al., 2022; Cheng et al., 2023) propose to extract fundamental symmetries of dynamics through an infinite-dimensional linear operator or revealing the time-reversal symmetry (T-symmetry) in TSRL Cheng et al. (2023). However, these methods are built on the downstream RL algorithm based on the action-level constraint and can still suffer from the over-conservatism problem that results in performance degeneration in small data settings. By contrast, TELS incorporates a T-symmetry regularized guide-policy to output the next valuable state in the learned latent space, which bypasses the conservative action-level behavioral constraint approaches.

## 6 DISCUSSION AND CONCLUSION

In this paper, we propose a highly sample-efficient offline RL algorithm that learns optimized policy within the compact informative latent space regulated by the T-symmetry in the dynamical systems. Specifically, we develop a T-symmetry enforced inverse dynamics model (TS-IDM) to construct a representative and generalizable latent space, effectively mitigating the challenges of OOD generalization. By integrating a T-symmetry-regularized guide-policy within this latent space, we can derive the valuable and reasonable next state for the learned latent inverse dynamics model to generate the optimized action for control, which leads to an impressively strong performance. Moreover, TS-IDM can simply function as a representation model to provide informative representations and improve the performance of other methods under the small-sample setting. From extensive experiments, we demonstrated that TELS essentially achieves the SOTA performance under the small-sample D4RL dataset. One limitation of TELS is that in complex, high-dimensional scenarios with limited datasets, extracting fundamental features becomes increasingly difficult. As part of future work, we aim to explore more expressive models to better capture these fundamental patterns.

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

# A    IMPLEMENTATION DETAILS

**Implementation details TS-IDM.**    We now present the implementation details of TS-IDM. We list the hyperparameters details of TS-IDM in Table 4, including the details of the structure we have implemented as well as the hyperparameters we used for training the TS-IDM.

We follow the same approach of TSRL (Cheng et al., 2023) that calculates the ODEs over the state for the forward $h_{fwd}$ and reverse $h_{rvs}$. Specifically, we calculate the time-derivative of the state encoder $\frac{d\phi_s(s)}{dt}$ and decoder $\frac{d\psi_s(z_s)}{dt}$ by computing their the jacobian matrix through vmap() function in Functorch.[1]

We have provided the learning curves of TS-IDM in Appendix C. Notably, our proposed TS-IDM proves to be easier to train compared to the T-symmetry enforced dynamics model (TDM) introduced in TSRL Cheng et al. (2023). In TSRL, one has to pretrain the encoder and decoders for several epochs before jointly training the model components to alleviate the learning instability. In contrast, TS-IDM does not require any pertaining to stabilize the training process. All components of TS-IDM can be jointly trained in a single stage, due to the strong consistency among its internal components.

Table 4: Hyperparameters of TS-IDM.

|  | Hyperparameters | Value |
|---|---|---|
| TS-IDM Architecture | State encoder hidden units | $512 \times 256$ |
|  | State encoder activation function | ReLU |
|  | Latent forward model hidden units | $256 \times 256$ |
|  | Latent forward model activation function | ReLU |
|  | Latent reverse model hidden units | $256 \times 256$ |
|  | Latent reverse model activation function | ReLU |
|  | latent inverse model hidden units | $1024 \times 1024 \times 1024$ |
|  | Latent inverse model activation function | ReLU |
|  | Latent inverse model dropout | True |
|  | Latent inverse model dropout rate | 0.1 |
|  | State decoder hidden units | $256 \times 512$ |
|  | State decoder activation function | ReLU |
|  | Action decoder hidden units | $512 \times 512$ |
|  | Action decoder activation function | ReLU |
| Hyperparameters | Optimizer type | Adam |
|  | $\beta$ | 1 (locomotion tasks); 0.1 (antmaze tasks) |
|  | Weight of $\ell_{s-rec}$ | 1 |
|  | Learning rate | 3e-4 |
|  | Batch size | 256 |
|  | Training epoch | 1000 |
|  | State normalize | True |
|  | Weight decay | 0 (locomotion tasks); 1e-5 (antmaze tasks and full dataset setting) |

**Implementation details for T-symmetry regularized guide-policy optimization.**    We present the structure details of the value functions and guide-policy in the following Table 5, we also list the training hyperparameter usage of all the small-sample experiments in Table 6.

Table 5: Structure and training parameters of guide-policy optimization

|  | **Hyperparameters** | **Value** |
|---|---|---|
| Guide-policy structure | Value network hidden units | $1024 \times 1024$ |
|  | Value network activation function | ReLU |
|  | Policy network hidden units | $1024 \times 1024$ |
|  | Policy network hidden units | ReLU |
| Training Perparameters | Optimizer type | Adam |
|  | Target Value network moving average | 0.05 |
|  | Batch size | 256 |
|  | Training steps | 100000 |
|  | State normalize | True |

---

[1]https://pytorch.org/functorch/stable/functorch.html

Table 6: Hyperparameters of guide-policy optimization used in experiments.

| Task | Guide-policy learning objective | $\tau$ | $\alpha$ | $\eta$ | dropout | Policy&Value learning rate |
|------|-------------------------------|--------|----------|--------|---------|----------------------------|
| hopper-medium-v2 | (11) | 0.7 | 0.1 | 10 | False | 1e-4 |
| halfcheetah-medium-v2 | (11) | 0.5 | 0.01 | 5 | True | 1e-4 |
| walker2d-medium-v2 | (11) | 0.5 | 0.01 | 5 | True | 1e-4 |
| hopper-medium-replay-v2 | (11) | 0.5 | 0.01 | 5 | False | 1e-5 |
| halfcheetah-medium-replay-v2 | (11) | 0.5 | 0.01 | 5 | True | 1e-4 |
| walker2d-medium-replay-v2 | (11) | 0.5 | 0.01 | 5 | True | 1e-4 |
| hopper-medium-expert-v2 | (11) | 0.7 | 0.1 | 10 | False | 1e-4 |
| halfcheetah-medium-expert-v2 | (11) | 0.5 | 0.01 | 5 | True | 1e-4 |
| walker2d-medium-expert-v2 | (11) | 0.5 | 0.01 | 5 | True | 1e-4 |
| all antmaze tasks | (12) | 0.9 | 10 | / | True | 1e-3 |

Table 7: Average normalized scores on full D4RL datasets of MuJoCo and Antmaze tasks.

| Task | BC | TD3+BC | CQL | IQL | DOGE | POR | TSRL | TELS (ours) |
|------|-----|--------|-----|-----|------|-----|------|-------------|
| Hopper-m | 52.9 | 59.3 | 58.5 | 66.3 | **98.6 $\pm$ 2.1** | 78.6 $\pm$ 7.2 | 86.7$\pm$8.7 | 94.3 $\pm$ 2.8 |
| Hopper-mr | 18.1 | 60.9 | 95.0 | 94.7 | 76.2$\pm$17.7 | 98.9 $\pm$ 2.1 | 78.7$\pm$28.1 | **99.5 $\pm$ 2.3** |
| Hopper-me | 52.5 | 98.0 | **105.4** | 91.5 | 102.7$\pm$ 5.2 | 90.0 $\pm$ 12.1 | 95.9$\pm$18.4 | **105.4 $\pm$ 8.5** |
| Halfcheetah-m | 42.6 | **48.3** | 44.0 | 47.4 | 45.3$\pm$ 0.6 | 48.8 $\pm$ 0.5 | 48.2 $\pm$0.7 | 44.3 $\pm$ 0.4 |
| Halfcheetah-mr | **55.2** | 44.6 | 45.5 | 44.2 | 42.8 $\pm$0.6 | 43.5$\pm$0.9 | 42.2$\pm$3.5 | 41.1 $\pm$ 0.1 |
| Halfcheetah-me | 55.2 | 90.7 | 91.6 | 86.7 | 78.7$\pm$8.4 | 94.7$\pm$2.2 | **92.0$\pm$1.6** | 87.1 $\pm$ 2.9 |
| Walker2d-m | 75.3 | 83.7 | 72.5 | 78.3 | **86.8 $\pm$ 0.8** | 81.1 $\pm$ 2.3 | 77.5 $\pm$4.5 | 81.3$\pm$ 5.1 |
| Walker2d-mr | 26.0 | 81.8 | 77.2 | 73.9 | **87.3 $\pm$ 2.3** | 76.6 $\pm$ 6.9 | 66.1$\pm$12.0 | 86.0 $\pm$ 3.3 |
| Walker2d-me | 107.5 | 110.1 | 108.8 | 109.6 | 110.4$\pm$1.5 | 109.1 $\pm$ 0.7 | 109.8$\pm$3.12 | **110.7 $\pm$ 1.4** |
| Antmaze-u | 65.0 | 78.6 | 84.8 | 85.5 | **97.0 $\pm$ 1.8** | 90.6 $\pm$ 7.1 | 81.4 $\pm$ 19.2 | 94.5 $\pm$ 10.3 |
| Antmaze-u-d | 45.6 | 71.4 | 43.4 | 66.7 | 63.5 $\pm$ 9.3 | 71.3 $\pm$ 12.1 | 76.5 $\pm$ 29.7 | **79.7 $\pm$ 15.3** |
| Antmaze-m-d | 0.0 | 0.0 | 54.0$\pm$11.7 | 74.6$\pm$3.2 | 77.6$\pm$6.1 | 79.2$\pm$3.1 | 0.0 | **82.4 $\pm$ 4.5** |
| Antmaze-m-p | 0.0 | 0.0 | 65.2$\pm$4.8 | 70.4$\pm$5.3 | 80.6$\pm$6.5 | 84.6 $\pm$5.6 | 0.0 | **86.7 $\pm$ 5.7** |
| Antmaze-l-d | 0.0 | 0.0 | 31.6$\pm$9.5 | 45.6$\pm$7.6 | 36.4 $\pm$9.1 | **73.4 $\pm$8.5** | 0.0 | 41.7 $\pm$ 14.2 |
| Antmaze-l-p | 0.0 | 0.0 | 18.8$\pm$15.3 | 43.5$\pm$4.5 | 48.2$\pm$8.1 | 58.0 $\pm$ 12.4 | 0.0 | **60.7 $\pm$ 13.3** |

**Reduced-size dataset generation.** To create reasonable reduced-size D4RL datasets for a fair comparison, we borrow a similar approach from the TSRL paper (Cheng et al., 2023) by randomly sub-sampling trajectories to construct the small dataset for training.

# B  ADDITIONAL RESULTS

## B.1  EVALUATION ON THE FULL DATASETS

We also evaluate the performance of TELS on the original full datasets of D4RL tasks, the results are presented in Table 7. Our proposed method achieves comparable or better performance as compared to existing offline RL methods. Moreover, we notice that with larger data size and broader state-action space coverage, the strong T-symmetry regularization in the TS-IDM can be properly relaxed, as sufficient data samples can be used to learn the model reasonably well. Therefore, we can trade off some regularization to promote model expressiveness (i.e., lower model learning loss). Specifically, for Antmaze tasks with the full dataset, we set the regularization hyperparameter $\beta = 0.01$ to train the TS-IDM. We also provide additional ablation experiments on the influence of the hyperparameter $\beta$ in the following section.

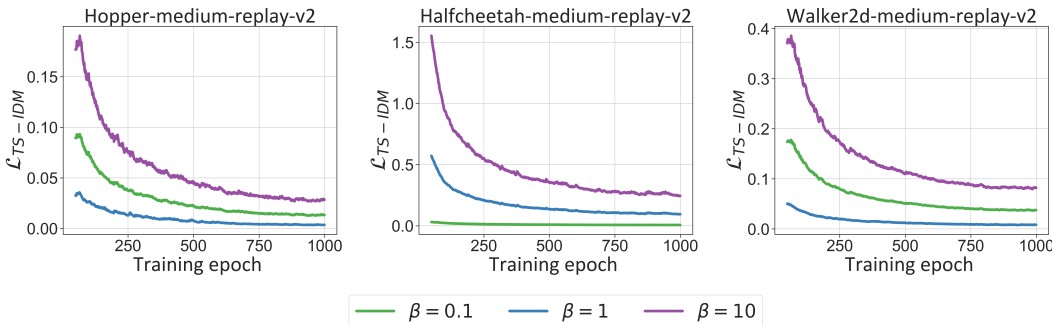

Figure 3: The learning curves for training TS-IDM on 10k dataset with different $\beta$ hyperparameter.

Table 8: Performance of TELS on 10k D4RL MuJoCo datasets when using TS-IDM with different $\beta$ hyperparameters.

|  | $\beta = 10$ | $\beta = 1$ | $\beta = 0.1$ |
|---|---|---|---|
| Hopper-m | $77.3 \pm 5.4$ | $\mathbf{77.3 \pm 10.7}$ | $61.4 \pm 5.6$ |
| Hopper-mr | $15.3 \pm 6.6$ | $\mathbf{43.2 \pm 3.5}$ | $19.7 \pm 3.4$ |
| Hopper-me | $37.6 \pm 17.9$ | $\mathbf{100.9 \pm 6.8}$ | $64.7 \pm 3.3$ |
| Halfcheetah-m | $32.9 \pm 2.3$ | $40.8 \pm 0.6$ | $\mathbf{41.2 \pm 1.1}$ |
| Halfcheetah-mr | $8.6 \pm 1.8$ | $33.2 \pm 1.0$ | $\mathbf{34.0 \pm 2.2}$ |
| Halfcheetah-me | $7.5 \pm 2.2$ | $\mathbf{40.7 \pm 1.2}$ | $39.5 \pm 2.1$ |
| Walker2d-m | $37.2 \pm 7.9$ | $\mathbf{62.4 \pm 5.3}$ | $54.6 \pm 8.2$ |
| Walker2d-mr | $17.1 \pm 2.9$ | $\mathbf{54.8 \pm 6.0}$ | $39.2 \pm 8.6$ |
| Walker2d-me | $20.4 \pm 10.4$ | $\mathbf{87.4 \pm 13.3}$ | $44.7 \pm 9.8$ |

## B.2 ADDITIONAL ABLATION EXPERIMENTS

**Impact of T-symmetry regularization on TS-IDM.** To investigate the impact of T-symmetry regularization strength controlled by the hyperparameter $\beta$ in Eq. (9), we conduct additional ablation experiments by varying the value of $\beta$ to assess how T-symmetry regularization influences the representation learning quality and downstream policy's performance. Specifically, we train TS-IDM on reduced-size 10k D4RL MuJoCo datasets with $\beta = \{0.1, 1, 10\}$, representing different T-symmetry regularization strengths. The learning curves of TS-IDM's overall learning loss $\mathcal{L}_{TS-IDM}$ in Eq. (9) are presented in Figure 3, and the final policy learning performances with different TS-IDM models are presented in Table 8.

From Figure 3, we can observe that choosing a proper $\beta$ value impacts the learning quality of TS-IDM. A large $\beta$ (e.g., $\beta = 10$) could impose overly strong regularization and hurt model expressiveness, which is reflected in the high learning loss at convergence. However, when the regularization strength is lowered, maintaining a proper scale of $\beta$ is important to ensure both the quality and generalizability of the learned representations. As we can see in Figure 3, in the Hopper and Walker2d tasks, choosing $\beta = 1$ provides the lowest $\mathcal{L}_{TS-IDM}$ loss; whereas in the Halfcheetah task, $\mathcal{L}_{TS-IDM}$ is the lowest when choosing $\beta = 0.1$. If we check the final policy's performance under different TS-IDMs in Table 8, we can see a clear correlation with what we have observed in Figure 3. TELS achieves the highest score on Hopper and Walker2d tasks when $\beta = 1$, but the scores are higher for Halfcheetah tasks when $\beta = 0.1$. This matches exactly with the learning performance of TS-IDM under different $\beta$ values. The strong correlation between TS-IDM's learning performance and the final policy's performance of TELS shows that we can select the best $\beta$ hyperparameter values by simply looking at TS-IDM's training loss, and using the one that provides the lowest training loss. This avoids the need to perform potentially unsafe online policy evaluations or unstable offline policy evaluations, which is favorable in real-world deployments.

**Additional ablations on the quality of learned representations.** To evaluate the quality of the representations learned by TS-IDM, we conduct comparison experiments by integrating the

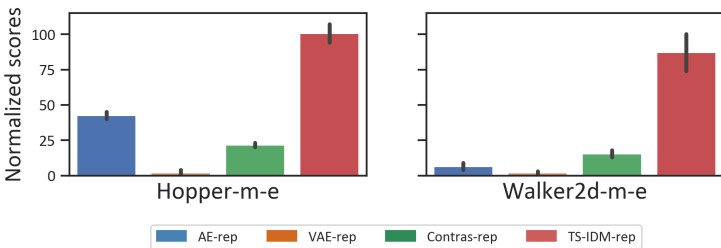

Figure 4: The performance of TELS with different representation models on 10k datasets, error bars indicate min and max normalized scores over 5 random seeds.

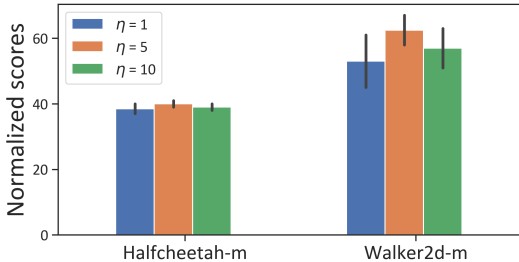

Figure 5: The performance of TELS with various weight $\eta$, error bars indicate min and max normalized scores over 5 random seeds.

downstream policy optimization process of TELS with various representation models, including an autoencoder("AE-rep"), a variational autoencoder("VAE-rep"), a contrastive representation model("Contras-rep"), and our proposed TS-IDM. The results demonstrate that with only 10k available samples, the TS-IDM representation achieves the best performance than other models. Notablely, "VAE-rep" derives the lowest performance, as it struggles to learn meaningful latent representations while simultaneously satisfying the prior distribution constraint in the extremely small dataset. While the "AE-rep" and "Contras-rep" models can achieve some performance, they fail to construct a generalizable and reliable latent space. In contrast, TS-IDM effectively provides reasonable and generalizable latent space, significantly enhancing performance in the reduced dataset setting.

**Ablations on hyperparameter $\eta$ on guide-policy optimization.** We also evaluated the performance of TELS with various $\eta = \{1, 5, 10\}$ under 10k dataset setting to examine its sensitivity on the state-level BC constraint in Eq. (11). This term regularizes the next state output by the guide-policy to align with the ground truth next state $s'$. We present the comparison results in Figure 5. The results show that TELS is generally robust with different $\eta$ values, but one needs to choose a proper scale of $\eta$ to ensure the best possible performance. When $\eta = 1$, the guide-policy $\pi_g$ may produce unrealistic next latent states, resulting in potential performance degradation. Conversely, a large $\eta$ ensures the output state remains close to the dataset, but this can be over-conservative, limiting the generalizability of the learned guide-policy.

**Ablations on the impact of T-symmetry consistency regularization on guide-policy optimization.** We also conduct ablation experiments in Table 9 to validate the effectiveness of the T-symmetry consistency regularizer term $\ell_{T\text{-}sym}(z_s, h_{inv}(z_s, \pi_g(z_s)))$ in Eq. (11) and (12). Specifically, we conduct ablation experiments on the reduced datasets of "Hopper-medium-v2", "Hopper-medium-expert-v2", "Walker2d-medium-v2", and "Walker2d-medium-expert-v2" tasks, each with 10k samples. The results demonstrate that incorporating the T-symmetry inconsistency term significantly enhances performance while reducing variance, indicating the importance and effectiveness of T-symmetry consistency regularization in small-sample offline policy learning.

### B.3 GENERALIZABILITY EVALUATION

To further investigate the generalizability of TELS, we construct a more challenging scenario by removing a portion of the critical data points from the "Antmaze-medium-diverse-v2" dataset, as

Table 9: Ablation experiments on the impact of $\ell_{T\text{-}sym}$ in TELS.

|  | Hopper-m | Hopper-m-e | Walker2d-m | Walker2d-m-e |
|---|---|---|---|---|
| w/o $\ell_{T\text{-}sym}$ | $62.1 \pm 15.1$ | $73.4 \pm 23.7$ | $54.6 \pm 8.2$ | $68.7 \pm 18.8$ |
| with $\ell_{T\text{-}sym}$ | $\mathbf{77.3 \pm 10.7}$ (↑24%) | $\mathbf{100.9 \pm 6.8}$ (↑51%) | $\mathbf{62.4 \pm 5.3}$ (↑15%) | $\mathbf{87.4 \pm 13.3}$ (↑26%) |

illustrated in the Figure 6 **(a)**. Specifically, we remove a proportion of data points in several critical regions along the paths from the start point to the goal-state location within the 0.1M dataset. We then train IQL, DOGE, POR and TELS on the remaining datasets to evaluate their performance under this extremely challenging scenario. We present the evaluation details in Figure 6.

The results demonstrate that all other baseline methods fail badly from 70% of data samples being deleted; only our method can still maintain reasonable performance. Moreover, in the 100% deletion ratio case, we can observe that only very sparse data samples are distributed in the boundaries of these critical regions, but with completely no information in the interior. However, our proposed TELS can still achieve reasonable performance even in this extremely challenging setting, by fully utilizing the limited information from the sparse data samples located in the boundaries of the data removal regions. These further demonstrate the strong generalization capability of our proposed method.

## C  LEARNING CURVES

The following are the learning curves of TS-IDM and the T-symmetry regularized guide-policy optimization in TELS on the reduced-size D4RL MuJoCo and Antmaze datasets. We evaluate the policy with 10 episodes over 5 random seeds.

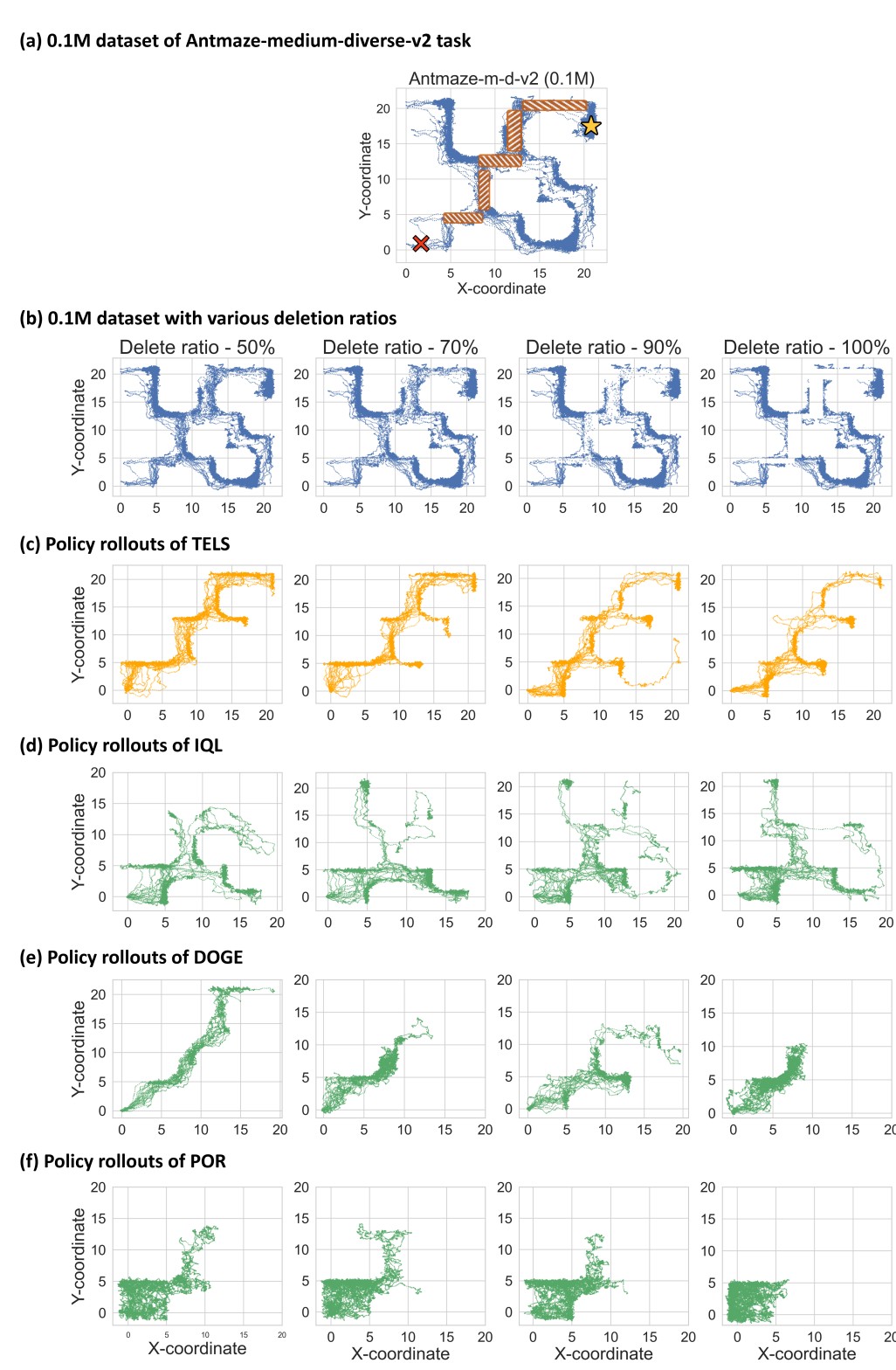

Figure 6: **(a)** Visualization of the states in "Antmaze-medium-diverse-v2" 0.1M dataset. The red cross and yellow star marks indicate the start and goal locations of the maze task. The deletion regions are marked as shaded brown rectangles. **(b-f)** Visualization of the trained policy rollouts for different algorithms across datasets with varying deletion ratios.

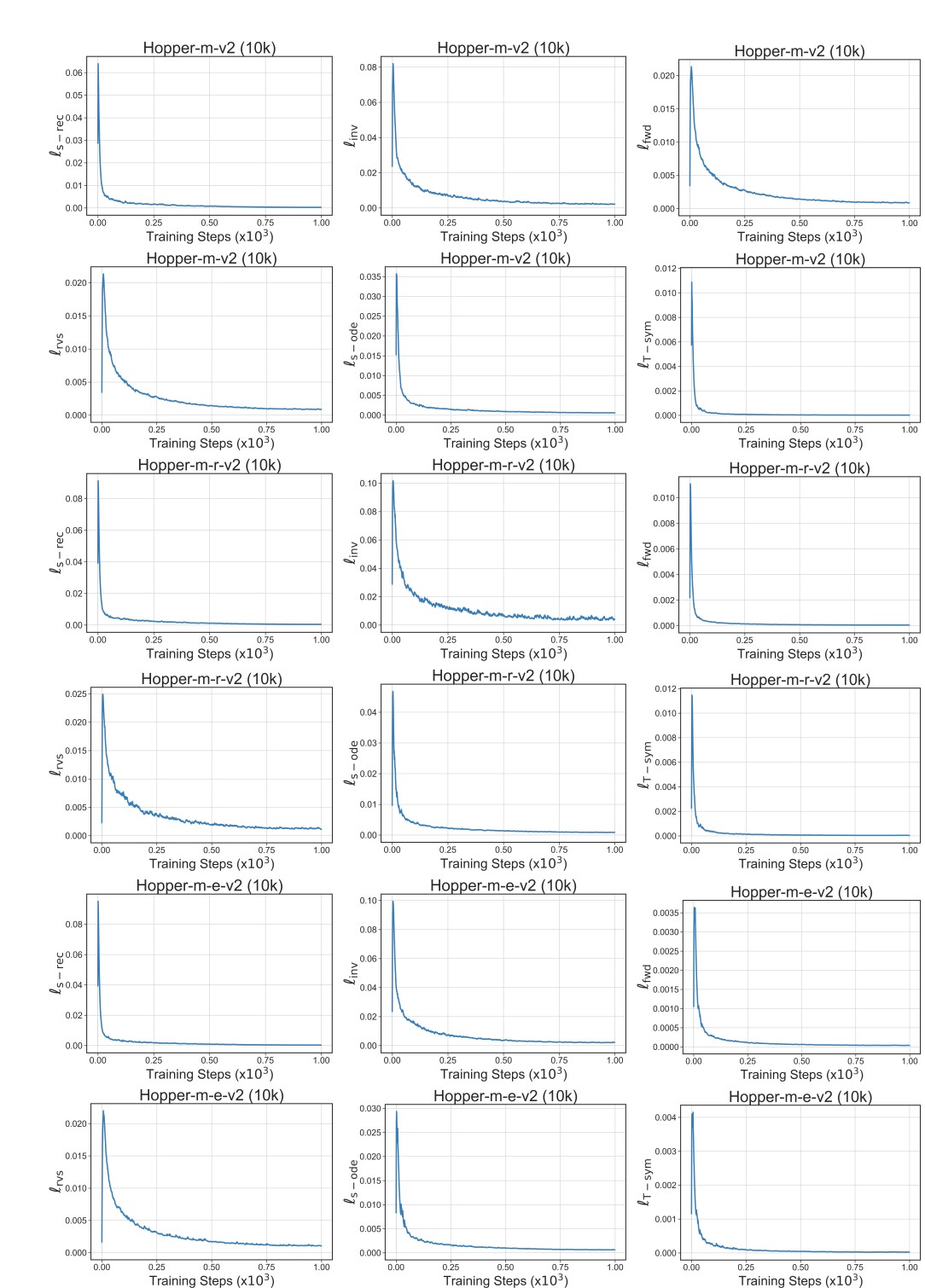

Figure 7: Learning curves of the overall and each individual loss terms in TS-IDM for Hopper tasks.

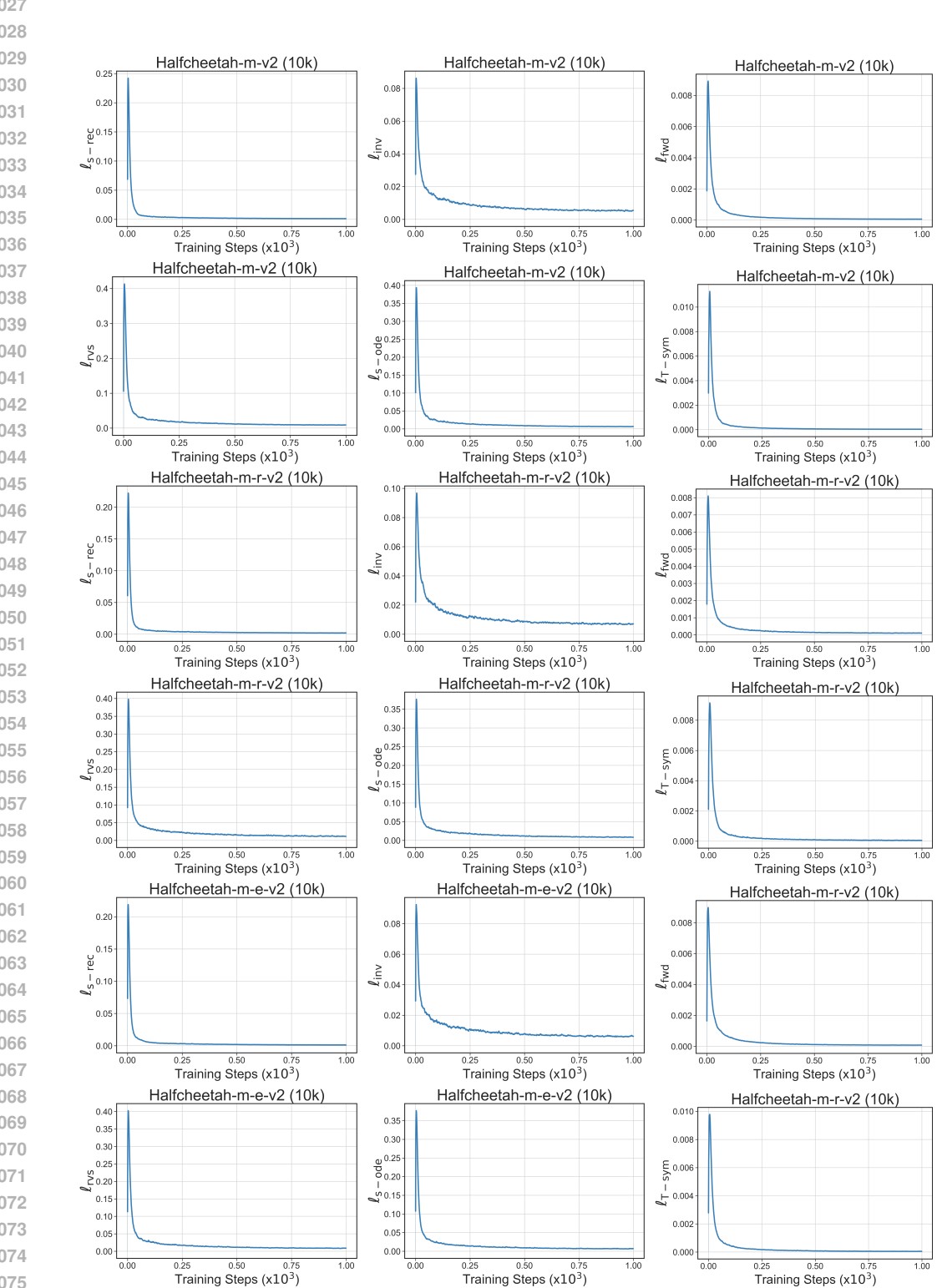

Figure 8: Learning curves of the overall and each individual loss terms in TS-IDM for Halfcheetah tasks.

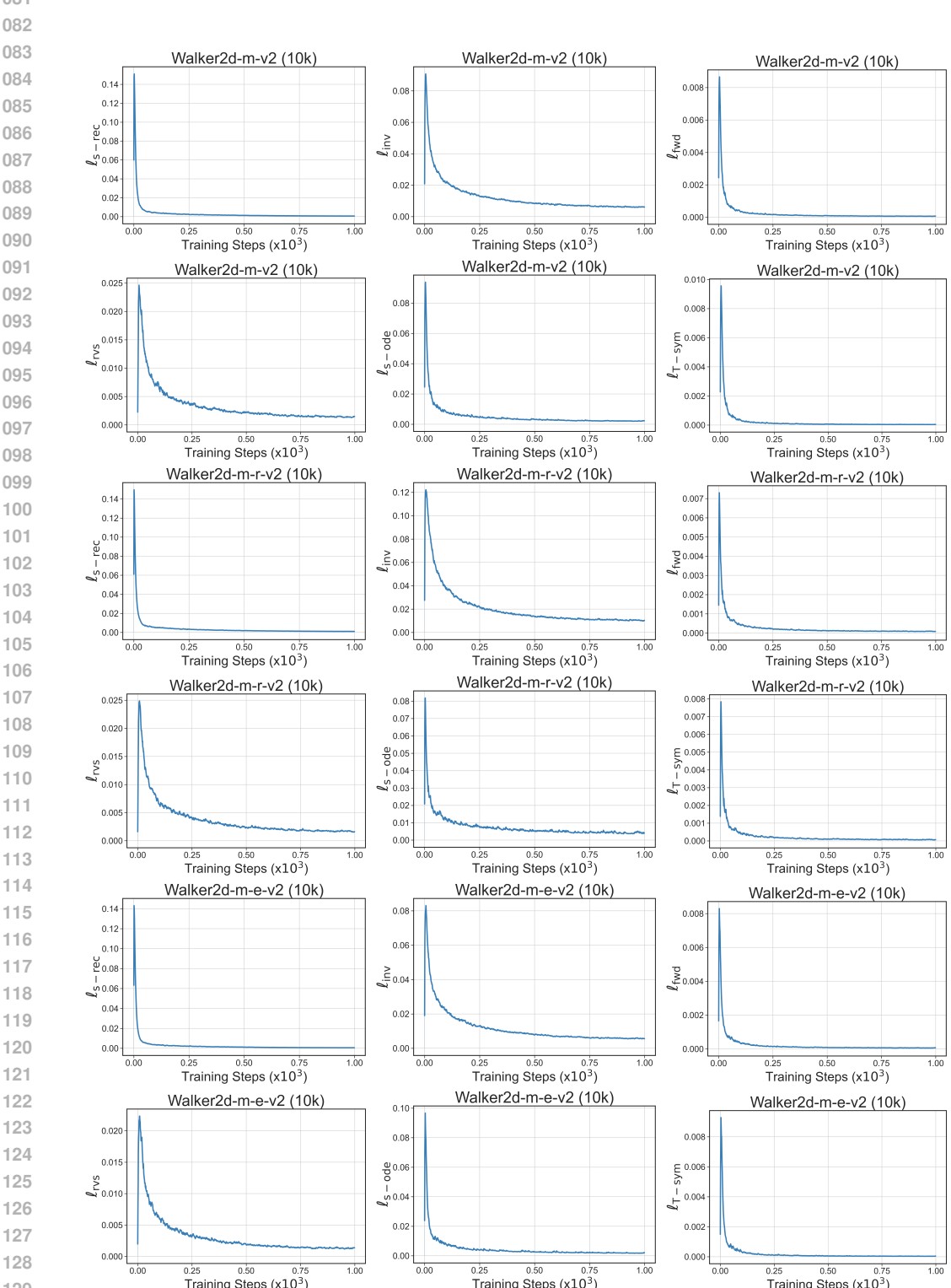

Figure 9: Learning curves of the overall and each individual loss terms in TS-IDM for Walker2d tasks.

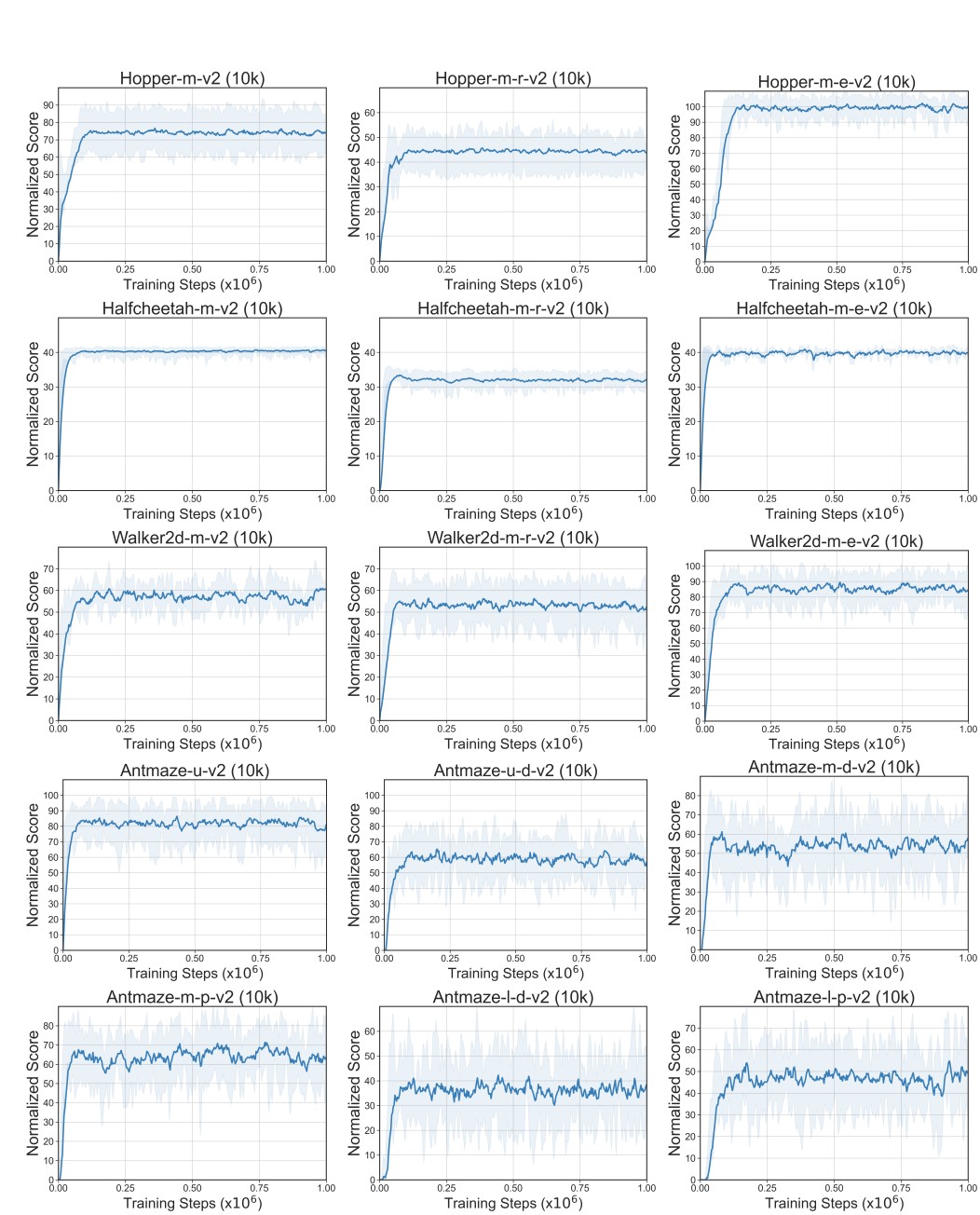

Figure 10: Learning curves of policy optimization in TELS for D4RL MuJoCo and Antmaze tasks with reduced-size datasets. We evaluate the policy within 10 episodes over 5 random seeds.

