# OpenReview forum: "Pushing the Limit of Small-Efficient Offline Reinforcement Learning"
_ICLR.cc/2025/Conference — Submitted to ICLR 2025_

### Official Review · Reviewer_dspK · 2024-10-24

**Soundness:** 3
**Presentation:** 3
**Contribution:** 2
**Rating:** 6
**Confidence:** 5

**Summary:**

This paper introduces a T-symmetry enforced inverse dynamics model (TS-IDM) into offline reinforcement learning. Overall the proposed TS-IDM is a latent space inverse dynamics model, which exploits the time-reversal symmetries of the underlying dynamics system to regularize the derived representations. The authors claim that such representations encode the information about the dynamics system, and generalize well in the out-of-distribution area. Based on this interpretation, they combined the representations with POR, a former offline RL  algorithm that stitches trajectories in the state space, and outperformed all existing baseline algorithms, especially in the low-data scenario.

**Strengths:**

1. The motivation is clearly presented. Sufficient dataset coverage is vital for reward maximization in both action-space or state-space and fine-grained representation may help to alleviate the need for dataset coverage.

2. The rationale behind the proposed method is plausible to me. I agree that the inductive bias of time-reversal symmetries extends beyond simple data augmentation techniques (e.g. Gaussian perturbations) and deserves more attention from the RL community. Although this is borrowed from TSRL, this paper refines several practices, such as maximizing the reward in the state space and thus constructing the latent representation based on states, rather than state-action pairs. The empirical performance looks good to me.

**Weaknesses:**

1. The overall objective looks extremely complex. It consists of 6 terms in total and each term is coupled with each other. Apart from the TS-IDM training, the policy optimization objective is also heavily dependent on hyper-parameters such as  $\tau$, $\alpha$, $\beta$, dropout rate and learning rates.

2. There is no clear discussion about the potential limitations of the proposed method. For example, the time-reversal symmetries seem to require that the dynamics can be expressed as an ODE. However, such ODE formulation may no apply to certain types of control systems, e.g. systems with stochastic transitions or partial observability. The authors are encourages to include more discussions about whether the ODE formulation is universally applicable, and if it is not, what is the scope of its application.

**Questions:**

1. The objective defined in Eq. 9 provides gradients for all modules in the TS-IDM, right? Did the authors detach the gradient of some parts of the TS-IDM training for some components of the loss?

2. Could the authors include performance and loss curves as supplementary materials? This would help readers comprehensively evaluate the proposed method from other perspectives, including learning dynamics, convergence speed, training stability, and also asymptotic performance.

some copy-edits:
1. line 218, $\phi_s$ should be $\psi_s$
2. line 651, TS-IMD should be TS-IDM
3. the loss term for inverse dynamics is missing in Eq. 9

---

> ### Author Response · Authors · 2024-11-23
> **Response to Reviewer dspK (Part 1/2)**
>
> We really appreciate the reviewer for the constructive comments and positive feedback on our paper. Regarding the concerns of the reviewer, we provide the following responses.
>
> > **W1. The overall objective looks extremely complex. It consists of 6 terms in total and each term is coupled with each other. Apart from the TS-IDM training, the policy optimization objective is also heavily dependent on hyper-parameters...**
>
> - We thank the reviewer for the thoughtful comment. Although the overall objective of TS-IDM looks complex, however, all of its internal components and loss terms are strongly coupled with each other, which allows them can be consistently learned without suffering from conflicts as observed in most multi-task learning problems. In fact, we find TS-IDM is very easy to train in our experiments. In all our MuJoCo locomotion tasks and antmaze tasks, we only use a single set of hyperparameters to train TS-IDM. We also provide all the learning curves of TS-IDM and our offline policy optimization procedure in Appendix C, in which the reviewer can clearly see its stable learning curves.
> - Moreover, we also find that our proposed TS-IDM is even easier to learn than the T-symmetry enforced dynamics model (TDM) proposed in TSRL [1]. In the TSRL paper, their authors mentioned that the TDM needs to first pre-train the encoder and decoders before joint training on other elements, otherwise will suffer from learning stability issues. However, in our proposed TS-IDM, we do not need to pre-train the encoder and decoders. All its components can be jointly learned in a single stage, due to its strong internal strong consistency among all of its components.
> - Regarding the dependency on the hyperparameters, we actually only tuned each hyperparameter from a very limited set of values during downstream policy learning, e.g., $\tau\in$ {0.5, 0.7} for MujoCo tasks, $\tau=0.9$ for all Antmaze tasks; $\eta=${5, 10} for MuJoCo tasks and $\eta=10$ for all Antmaze tasks, etc. We find in our experiments that using a single set of hyperparameters for TELS in many cases can already ensure reasonably good results, but slightly tuning the hyperparameter can achieve better performance. We have conducted additional ablation studies on the impacts of hyperparameters in both TS-IDM and policy optimization. Please check Appendix A and B.2 in our revised paper for more detailed results and discussions. We hope this can address some of the reviewer's concerns.
>
> [1] Cheng, P., et al. Look beneath the surface: Exploiting fundamental symmetry for sample-efficient offline RL. NeurIPS 2023.

---

> ### Author Response · Authors · 2024-11-23
> **Response to Reviewer dspK (Part 2/2)**
>
> **W2. There is no clear discussion about the potential limitations of the proposed method. For example, the time-reversal symmetries seem to require that the dynamics can be expressed as an ODE. However, such ODE formulation may no apply to certain types of control systems, e.g. systems with stochastic transitions or partial observability. The authors are encourages to include more discussions about whether the ODE formulation is universally applicable, and if it is not, what is the scope of its application.**
>
> - We thank the reviewer for this comment. The T-symmetry used in our method actually only requires the dynamics can be expressed as an ODE in the latent space, rather than in the original state-action space. Even if the dynamical system is nonlinear, we can still construct an ODE dynamical system in the latent space by applying a suitable mapping function, such as the learned encoder and decoder in TS-IDM. In fact, many works in control theory and physics follow a similar treatment, such as Koopman theory [2, 3] represents the arbitrary nonlinear dynamics in terms of an infinite-dimensional linear operator; discovering governing equations in physics (e.g., SinDy methods [4, 5]) also follows a similar recipe that uses encoder and decoder to map the original system into the latent space, where the dynamics become an ODE system. The universality of T-symmetry in RL modeling is extensively discussed in the TSRL paper [1]. As the use of T-symmetry in RL is not first proposed by us, hence we refer the readers to check the TSRL paper rather than providing an extensive discussion in our paper (see our discussion in the Preliminary section).
> - However, we do acknowledge there are some potential limitations of our method. The main limitations are the relatively complex model architecture and the additional hyperparameters as discussed in our response to W1. We have provided additional analyses and ablations in Appendix A and B.2 in our revised paper for more discussion.
>
>
> [2] Igor Mezic. Spectral properties of dynamical systems, model reduction and decompositions. Nonlinear Dynamics, 2005.
>
> [3] Matthias Weissenbacher, et al. Koopman q-learning: Offline reinforcement learning via symmetries of dynamics. ICML 2022.
>
> [4] Steven L Brunton, et al. Discovering governing equations from data by sparse identification of nonlinear dynamical systems. PNAS 2016.
>
> [5] Kathleen Champion, et al. Data-driven discovery of coordinates and governing equations. PNAS 2019.
>
> > **Q1. The objective defined in Eq. 9 provides gradients for all modules in the TS-IDM, right? Did the authors detach the gradient of some parts of the TS-IDM training for some components of the loss?**
>
> Yes, all loss terms are jointly optimized when learning TS-IDM. We do not detach some parts of TS-IDM during training. Note that this is needed for the IDM in TSRL, which needs to first detach other loss terms and pre-train the encoder and decoders, and then jointly train all elements together. However, we find that this is not needed for our proposed TS-IDM, as all elements and loss terms are strongly coupled with each other, hence they can be jointly trained and enjoy stable learning. Please check the added learning curves for TS-IDM in Appendix C of our revised paper, which provides a more clear illustration of the learning process for TS-IDM.
>
> > **Q2. Could the authors include performance and loss curves as supplementary materials? This would help readers comprehensively evaluate the proposed method from other perspectives, including learning dynamics, convergence speed, training stability, and also asymptotic performance.**
>
> We thank the reviewer for this constructive comment. We have included all the learning curves of TS-IDM and the policy learning process. Please check Appendix C for details.
>
> > **Typos**
>
> We thank the reviewer for these comments. We have revised these issues in our updated paper.

---

> > ### Comment · Reviewer_dspK · 2024-11-23
> >
> > Thanks for the reply. Most of my concerns are addressed, and I still encourage the authors to discuss the applicability of T-symmetry in the final version of the paper. One more comment on the hyper-parameters: apart from the hyper-parameters that the authors noted in the response ($\eta, \tau, \beta$), it can be noticed from Table 6 that the authors also tuned the dropout, $\alpha$, and learning rate for each dataset. As far as I know, this is beyond the typical budget for hyper-parameter tuning in the field of offline RL.

---

> > > ### Author Response · Authors · 2024-11-25
> > > **Thanks for your feedback!**
> > >
> > > We sincerely appreciate the reviewer for the valuable feedback!
> > >
> > > **Regarding the suggestion of adding discussion on T-symmetry:**
> > > - We will follow the advice from the reviewer to add discussions on the wide applicability of T-symmetry in our final paper.
> > >
> > > **Regarding the remaining concern on hyper-parameter tuning:**
> > > - We wish to highlight that we use the same set of hyper-parameters for training all Halfcheetah and Walker2d tasks, as both of them share identical state and action dimensions (17 for states and 6 for actions). In contrast, Hopper tasks have a smaller state-action space (11 states and 3 actions), making it comparatively simpler given the same amount of training data (e.g., 10k samples). Consequently, we opted for a more aggressive learning strategy for Hopper tasks, i.e., using higher $\alpha=0.1$ to focus more on value maximization. We also adopt a similar logic to select the hyperparameters for training all Antmaze tasks, and all our results for Antmaze tasks are conducted using a single set of hyper-parameters  (please refer to Table 1 and 6 in our paper for details).
> > > - To further address the reviewer’s concern, we added additional experiments to evaluate the performance of TELS using the same set of hyper-parameters across all MuJoCo tasks. Specifically, we train TELS on Hopper tasks using **the same hyperparameter setting as those for Walker2d and Halfcheetah**. The results are reported in the following table.
> > >
> > > |  | $\tau$  | $\alpha$ | $\eta$ | Dropout | Policy & Value learning rate|
> > > | -------- |-------- | -------------- |----------- |----- |----- |
> > > | All mujoco tasks  | 0.5 | 0.01 |5 |True |1e-4 |
> > >
> > >
> > > | **Task** | BC| TD3+BC | CQL | IQL |DOGE| POR  | TSRL| TELS (using the same set of hyper-parameters) | TELS (reported in the paper)
> > > | ---------- |----------- | ----------- | ----------- | ----------- | ----------- | ----------- | ----------- | ----------- | ----------- |
> > > | Hopper-m   | 29.9 $\pm$  11.7 | 40.1 $\pm$ 18.6 | 43.1 $\pm$  24.6| 46.7 $\pm$ 6.5 | 44.2 $\pm$ 10.2    |46.4 $\pm$ 1.7 |62.0 $\pm$ 3.7  | 65.5 $\pm$ 12.4 | 77.3 $\pm$ 10.7
> > > | Hopper-m-r   | 12.1 $\pm$ 5.3 | 7.3 $\pm$ 6.1 | 2.3 $\pm$ 1.9 | 13.4 $\pm$ 3.1 | 17.9 $\pm$ 4.5 | 17.4 $\pm$ 6.2   | 21.8 $\pm$ 8.2 | 30.1 $\pm$ 7.6 | 43.2 $\pm$ 3.5
> > > | Hopper-m-e  |27.8 $\pm$ 10.7 | 17.8 $\pm$ 7.9 | 29.9 $\pm$ 4.5 | 34.3 $\pm$ 8.7 | 50.5 $\pm$ 25.2 | 37.9 $\pm$ 6.1  |50.9 $\pm$ 8.6  | 71.5 $\pm$ 9.1 | 100.9 $\pm$ 6.8
> > >
> > > As the reviewer can observe from the above results, even using a single set of hyper-parameters, TELS still achieves much better performance as compared to all other offline RL baseline methods. In practice, for tasks with relatively abundant offline data and higher state-action space coverage, we suggest the practitioner can reasonably relax the regularization strength (e.g., increase $\alpha$) to enable even better performance of TELS. But even using the same set of hyper-parameters, as we have shown in our results, TELS can already achieve good performance that outperforms existing baseline methods.

---

> > > > ### Comment · Reviewer_dspK · 2024-11-25
> > > >
> > > > Thanks for the additional results. Based on my current evaluation and the discussion, I will maintain my score and increase my confidence.

---

### Official Review · Reviewer_3YZb · 2024-10-30

**Soundness:** 2
**Presentation:** 3
**Contribution:** 1
**Rating:** 3
**Confidence:** 4

**Summary:**

This paper first proposes a T-symmetry enforced inverse dynamics model (TS-IDM) to construct a latent space. And then use the T-symmetry to regularize the guide-policy learning. The method TELS (Offline RL via T-symmetry Enforced Latent State-Stitching) exhibits exceptionally high data efficiency on D4RL with small-sample setting.

**Strengths:**

1. The paper provides a thorough introduction to the background and preliminaries. Combined with the clear writing logic, the entire paper is quite easy to understand.
2. The proposed method has achieved a performance that significantly surpasses all baselines under the small-sample setting, reaching the current state-of-the-art (SOTA).

**Weaknesses:**

The paper seems to be a **straightforward fusion** of POR (Policy-guided Offline RL) and TSRL (T-Symmetry regularized offline RL).

1. Compared with TDM in TSRL, the proposed TS-IDM introduces a latent inverse dynamics model ($h_{inv}$), which appears to be designed for the policy execution in POR.
2. While TSRL employs the offline algorithm TD3+BC, this paper chooses POR.

In summary, I believe the novelty and contribution of the proposed algorithm are relatively limited.

**Questions:**

1. In overall learning objective (9), the loss function of inverse dynamic model (4) is missing.
2. Considering TS-IDM as a specialized form of dynamic model, it is essential to compare it with model-based offline algorithms.
3. On line 361, the sentence is missing a period at the end.

---

> ### Author Response · Authors · 2024-11-23
> **Response to Reviewer 3YZb (Part 1/2)**
>
> We thank the reviewer for the comments. Regarding the concerns of the reviewer, we provide the following responses.
>
> > **W1. The paper seems to be a straightforward fusion of POR (Policy-guided Offline RL) and TSRL (T-Symmetry regularized offline RL).**
>
> We'd like to clarify that our proposed method is not a straightforward fusion of TSRL and POR, but is specifically designed to tackle the small-sample learning challenge in offline RL. Notably, there are many key differences of our method as compared to TSRL and POR:
>
> First, if the reviewer checks the TSRL paper, it is easy to find that the **architecture and designs of the T-symmetry enforced dynamics model (TDM) in TSRL is very different from our proposed T-symmetry enforced inverse dynamics model (TS-IDM)**.
> - TDM is a reconstruction model to encode and decode both states and actions, whereas TS-IDM is overall an inverse dynamics model, which only encodes states, but decodes both states and actions. As we have discussed in Lines 142-152 in our paper, the design of TDM in TSRL has many drawbacks:
>     1. The only useful part of the learned TDM for downstream policy learning is its encoder $\phi(s, a)$, wasting the dynamics-related information captured in the model. By contrast, TS-IDM can be reused as an execute-policy to extract optimal actions.
>     2. TDM needs to access both the state and action to derive latent representations, making Q-function maximization the only option for policy optimization, which inevitably requires adding conservative action-level constraints.
>     3. Involving action as an input for representation learning is also prone to capturing the biased behaviors in the data-generating policy (e.g., data generated from expert policy will have special action patterns), which could impede learning fundamental, distribution-agnostic dynamics patterns in data. By contrast, TS-IDM only learns cleaner state representations with its state encoder $\phi_s(s)$, bypassing all these issues.
> - Also, the detailed learning procedure of TS-IDM of TELS is also different from IDM in TSRL:
>     1. TDM only enforces the ODE property for its encoder, but not the decoder. But TS-IDM enforces ODE property for both its state encoder and decoders. We introduce the loss term $\ell_{s-ode}$ in Eq. (7) specifically to achieve this goal, which is not considered in TSRL. We find that this design is actually very important as it can greatly enhance the coupling among the different elements in the model, and results in a more stable learning process. This is reflected in the different training procedures between TDM and TS-IDM. In the TSRL paper, one needs to first pre-train the TDM encoder and decoders before joint training on other elements, otherwise, TDM will suffer from learning stability issues. However, in our proposed TS-IDM, we do not need to pre-train the encoder and decoders. All its components can be jointly learned in a single stage.
>     2. In TDM of TSRL, its final training loss needs to add L1-norms of the parameters of the latent forward and reverse dynamics models to regularize their learning. However, this is not needed in our method (see Eq. (9)), as the design of our proposed TS-IDM enables strongly coupled and consistent relationships among all its internal components
>
> Second, **the downstream policy learning scheme of TELS also has several differences from TSRL and POR**:
> - TSRL uses TD3+BC as the backbone algorithm for policy learning, which suffers from the over-conservative action-level constraint in small dataset settings.
> - POR on the other hand, optimizes the policy on the original state space, which still needs the state-action space to have reasonable coverage to enable valid state-stitching. By contrast, TELS performs state-level policy optimization entirely in the generalizable and compact latent state space derived from the TS-IDM. Moreover, TELS adds T-symmetry consistency regularization in its guide-policy optimization objective, which is not considered in POR. It plays a critical role in preventing the guide-policy $\pi_g$ from outputting problematic and non-generalizable latent next state, thereby further enhancing logical OOD generalization.
>
> In summary, **all our design choices are made to maximally promote OOD generalization and small-sample performance of offline RL, rather than naively fusing existing algorithms.**

---

> > ### Comment · Reviewer_3YZb · 2024-11-28
> >
> > Thank authors for providing further clarification on the motivation behind the paper. The paper combines TSRL with POR. A point of discussion could be whether this fusion is straightforward. Although the authors have elaborated on the differences between this work and TSRL as well as POR, I still perceive the contribution to be limited.

---

> ### Author Response · Authors · 2024-11-23
> **Response to Reviewer 3YZb (Part 2/2)**
>
> > **Q1. In overall learning objective (9), the loss function of inverse dynamic model (4) is missing.**
>
> We sincerely apologize for the omission of $\ell_{inv}$ and the potential reader confusion. This loss term is jointly optimized with all other loss terms. We have added $\ell_{inv}$ in the complete training loss of TS-IDM (Eq. (9)) in our revised paper to correct this typo. We thank the reviewer for pointing out this issue.
>
> > **Q2. Considering TS-IDM as a specialized form of dynamic model, it is essential to compare it with model-based offline algorithms.**
>
> We'd like to clarify that our approach is very different from the model-based offline RL methods. Although we learn a T-symmetry enforced inverse dynamics model (TS-IDM), this model is **never used to generate model rollout data**. TS-IDM is only used to learn generalizable state representations and provide the T-symmetry consistency measures for downstream offline policy learning. This is completely different from the model-based RL, in which the models are used to generate synthetic rollout data during policy learning. If the reviewer checks our proposed algorithm, its design resembles more of a representation learning model with a model-free policy learning procedure. Moreover, as shown in the TSRL paper [1], model-based offline RL methods like MOPO [2] perform quite poor in the small dataset setting, as the limited sample cannot support learning a dynamics model with reasonable accuracy, and the high approximation error in the model rollout data severely deteriorate offline policy learning performance. Therefore, we focus more on comparing model-free offline RL baselines, which generally perform better in the small sample setting.
>
>
> > **Q3. On line 361, the sentence is missing a period at the end.**
>
> We thank the reviewer for the comment, we have revised it in our updated paper.
>
>
> **References**
>
> [1] Cheng, P., et al. Look beneath the surface: Exploiting fundamental symmetry for sample-efficient offline RL. NeurIPS 2023.
>
> [2] Yu, T., et al. MOPO: Model-based Offline Policy Optimization. NeurIPS 2020.

---

> ### Author Response · Authors · 2024-11-26
> **A gentle reminder for reviewer**
>
> Dear reviewer 3YZb,
>
> As the rebuttal phase is coming to a close, we wanted to check back to see whether you have any feedback or remaining questions. We would be happy to clarify further, and grateful for any other feedback you may provide.
>
> We really appreciate your time engaged in the review process and look forward to your replies!
>
> Best regards,
>
> Authors of Paper 10618

---

### Official Review · Reviewer_g1ie · 2024-10-31

**Soundness:** 3
**Presentation:** 3
**Contribution:** 3
**Rating:** 6
**Confidence:** 4

**Summary:**

The authors propose a new sample-efficient offline RL algorithm (TELS) based on a Time-Reversal Symmetry-Enforced Dynamics Model (TDM). Compared to the baseline algorithm (TSRL: Time-Reversal Symmetry Regularized Offline RL), they introduce a T-Symmetry Enforced Inverse Dynamics Model (TS-IDM) to derive regulated latent state representations that facilitate OOD generalization. Through state-stitching in a compact latent state space, they can efficiently obtain optimized next states from a guide policy. In experiments with small-sample tasks, they demonstrate superior performance compared to state-of-the-art offline RL algorithms.

**Strengths:**

* They address several weaknesses of existing algorithms by leveraging T-symmetry.
     * TELS separates the state-action encoder used in the TSRL algorithm into two components: a state encoder and a latent inverse dynamics module. By incorporating the state encoder in the policy optimization process, TELS promotes stable learning and improves the generalization of the state-value function (using an IQL-style method) through more compact and well-behaved representations.
     * Furthermore, by utilizing the latent inverse dynamics, the guide policy outputs the next states that maximize the reward. In action inference, no additional training process is required due to the use of the learned TS-IDM.
     * TSRL showed no noticeable performance improvement over existing offline algorithms, particularly in AntMaze. In contrast, TELS demonstrate significant performance improvements across most datasets.

**Weaknesses:**

* In Equation 9, the complete training loss function of TS-IDM comprises $\ell_{s-rec}$, $\ell_{fwd}$, $\ell_{rvs}$, $\ell_{s-ode}$ and $\ell_{T-sym}$. However, despite the latent inverse dynamics model's loss ($\ell_{inv}$) being specified in Equation 4, I am unable to locate this specific loss term. I am curious whether this loss is implicitly embedded within other terms or if it was simply not explicitly indicated.
*  In Figure 3, they examine the generalizability of TELS by constructing a more challenging scenario in which a portion of samples is removed. However, even as the deletion ratio increases, it appears that all state points are still represented in each figure, though the number of samples for each state decreases. It remains questionable whether performance would improve even if certain state areas were missing.

**Questions:**

In addition to questions about potential weaknesses, I would like to raise a few more questions:
* Compared between Table 1 (small data) and Table 7 (full data), Antmaze-m-p and Antmaze-l-p show slightly higher performance when using the smaller dataset. How would you interpret these results?
* In Table 3, IQL with TS-IDM representation shows improved performance (Hopper-m: 54.1, Walker2d-m: 41.3) but still underperforms compared to TELS (Hopper-m: 77.3, Walker2d-m: 62.4). Given that TELS utilizes an IQL-style state-value function update, the primary difference between TELS and IQL lies in the policy improvement or optimization process. What do you think is the key factor behind this result?
* In Algorithm 1 of the TSRL paper, they enhance policy training by adding augmented samples to the dataset. Are you proposing that your algorithm trains solely on the reduced data without any data augmentation?
* In Table 5 of the TSRL paper, they report results only for the Antmaze-u and Antmaze-u-d datasets. In contrast, in Table 1 of the TELS paper, TSRL results are included for additional datasets—such as Antmaze-m-d, m-p, l-d, and l-p. Since TELS performs well on these datasets, it seems unusual that TSRL, its baseline, shows a normalized score of zero on certain tasks. Does a zero score indicate that TSRL actually achieved no performance on these tasks, or does it mean these datasets were not evaluated?

---

> ### Author Response · Authors · 2024-11-23
> **Response to Reviewer g1ie (Part 1/2)**
>
> We appreciate the reviewer for the constructive comments on our paper, which are very helpful to improve the clarity of our paper. Regarding the concerns from the reviewer, we provide the detailed responses as follows:
>
> > **W1. ... despite the latent inverse dynamics model's loss ($\ell_{inv}$) being specified in Equation 4, I am unable to locate this specific loss term. I am curious whether this loss is implicitly embedded within other terms or if it was simply not explicitly indicated.**
>
> We sincerely apologize for the omission of $\ell_{inv}$ and the potential reader confusion. This loss term is jointly optimized with all other loss terms. We have added $\ell_{inv}$ in the complete training loss of TS-IDM (Eq. (9)) in our revised paper to correct this typo. We thank the reviewer for pointing out this issue.
>
> > **W2. In Figure 3, they examine the generalizability of TELS by constructing a more challenging scenario in which a portion of samples is removed. However, even as the deletion ratio increases, it appears that all state points are still represented in each figure, though the number of samples for each state decreases. It remains questionable whether performance would improve even if certain state areas were missing.**
>
> - We thank the reviewer for this comment. To further demonstrate the generalizability of TELS, we have added additional results in this experiment with 100% of the samples in the critical regions being removed, and also include comparisons with other baseline methods (i.e., IQL, DOGE, and POR). The results are presented in Appendix B.3 and Figure 6 in our revised paper.
> - If the reviewer checks our updated results, all other baseline methods start to fail when 70% of data samples are deleted, only our method can still maintain reasonable performance. Moreover, in the 100% deletion ratio case, only very sparse data samples remain in the boundaries of these critical regions, but with completely no information in the interior. However, our proposed TELS can still achieve reasonable performance even in this extremely challenging setting, by fully utilizing the limited information from the sparse data samples located in the boundaries of the data removal regions. These further demonstrate the strong generalization capability of our proposed method.
>
> > **Q1. Compared between Table 1 (small data) and Table 7 (full data), Antmaze-m-p and Antmaze-l-p show slightly higher performance when using the smaller dataset. How would you interpret these results?**
>
> - In our initial submission, we used the same regularization strength hyperparameter $\beta=0.1$ to train TS-IDM for both reduced-size and full datasets of Antmaze tasks for simplicity. However, we notice that this might pose too much regularization for the full dataset cases, as there are already sufficient data samples with high state-action space coverage to be used to learn the model reasonably well. Too much regularization might hurt model expressiveness, reflected in the increased training loss. Hence in our revised paper, we report the full dataset Antmaze tasks with $\beta=0.01$, under this case, TELS achieves much better performance. Please refer to our updated Appendix B.1 and Table 7 for detailed results and discussions.
> - Moreover, we also provide additional analysis to demonstrate the impact of $\beta$ for downstream offline policy learning, please see the "Impacts of T-symmetry regularization on TS-IDM" part in Appendix B.2 for detailed discussion. We find a strong correlation between TS-IDM's learning performance (lower training loss) and the improved policy performance of TELS. This shows that we can select the best $\beta$ hyperparameter values by simply looking at TS-IDM's training loss, without resorting to performing potentially unsafe online policy evaluations or complex offline policy evaluations.

---

> ### Author Response · Authors · 2024-11-23
> **Response to Reviewer g1ie (Part 2/2)**
>
> > **Q2. In Table 3, IQL with TS-IDM representation shows improved performance (Hopper-m: 54.1, Walker2d-m: 41.3) but still underperforms compared to TELS (Hopper-m: 77.3, Walker2d-m: 62.4). Given that TELS utilizes an IQL-style state-value function update, the primary difference between TELS and IQL lies in the policy improvement or optimization process. What do you think is the key factor behind this result?**
>
> The key reason that IQL with TS-IDM underperforms TELS is that IQL still adopts implicit action-level constraint for policy learning, whereas policy learning of TELS completely operates in the latent state space. In IQL, the use of expectile regression to learn action and state value functions essentially corresponds to an implicit regularization between policy $\pi(a|s)$ and the behavioral policy $\pi_{\beta}$, which still operate on the action level. We refer the reviewer to a recent paper [1] for a detailed discussion on this. By contrast, TELS performs offline policy optimization completely on the latent state space. It learns a guide-policy that outputs the optimal next latent state with completely no action-related constraint involved, and then uses the TS-IDM to extract the optimal action. Due to design, TELS enables state-stitching within latent space and achieves much better generalization performance.
>
>
> [1] Xu, et al. Offline RL with No OOD Actions: In-Sample Learning via Implicit Value Regularization. ICLR 2023.
>
> > **W3. In Algorithm 1 of the TSRL paper, they enhance policy training by adding augmented samples to the dataset. Are you proposing that your algorithm trains solely on the reduced data without any data augmentation?**
>
> We did not use the latent space data augmentation proposed by TSRL in our method. We actually tested it in our experiments and found its benefit is negligible, and sometimes could even have a negative impact. This is mainly because adding augmented data may introduce extra noise, which can often be detrimental to the small-sample learning setting. Moreover, tuning the scale of added noise is also quite tricky. Only utilizing TS-IDM for representation learning and T-symmetry consistency regularization, on the other hand, proved to be most effective based on our empirical observations. Therefore, to keep our algorithm clean and reduce unnecessary complexity, we did not use data augmentation in TELS.
>
> > **Q4. In Table 5 of the TSRL paper, they report results only for the Antmaze-u and Antmaze-u-d datasets. In contrast, in Table 1 of the TELS paper, TSRL results are included for additional datasets—such as Antmaze-m-d, m-p, l-d, and l-p. Since TELS performs well on these datasets, it seems unusual that TSRL, its baseline, shows a normalized score of zero on certain tasks. Does a zero score indicate that TSRL actually achieved no performance on these tasks, or does it mean these datasets were not evaluated?**
>
> Yes, TSRL fails in Antmze tasks with larger maps (e.g., medium and large maps). As the score of Antmaze tasks reflects the success rates in reaching the target location, the policy learned in TSRL never successfully reaches the destination under 0.1M training data. This is primarily because TSRL still uses TD3+BC as the backbone for policy learning, which is a very conservative algorithm with explicit action-level constraints. If the reviewer checks Table 1 and Table 7 of our paper, TD3+BC also fails to solve the Antmaze-m and -l tasks, which is consistent with the behavior of TSRL.

---

> > ### Comment · Reviewer_g1ie · 2024-11-25
> >
> > I appreciate the authors' detailed response to all my comments!
> >
> > As mentioned by Reviewer dspK, having many parameters to adjust could be considered a drawback. However, in my opinion, while parameters such as $\tau, \alpha, \eta$ and $\beta$ could be further optimized, it might be more appropriate to standardize settings for dropout and learning rate. Despite this, TELS demonstrates outstanding performance.
> >
> > So I raise my score!

---

> > > ### Author Response · Authors · 2024-11-25
> > > **Thank you for raising the score!**
> > >
> > > Dear Reviewer g1ie,
> > >
> > > We really appreciate your positive feedback on our paper! Regarding the additional suggestions on the hyper-parameters
> > >
> > > As in our response to Reviewer dspK, we conducted additional experiments on the Hopper tasks to use the same hyperparameter setting as those for Walker2d and Halfcheetah. The results are reported in the following table. We will also report these results in our final paper.
> > >
> > > |  | $\tau$  | $\alpha$ | $\eta$ | Dropout | Policy & Value learning rate|
> > > | -------- |-------- | -------------- |----------- |----- |----- |
> > > | All mujoco tasks  | 0.5 | 0.01 |5 |True |1e-4 |
> > >
> > >
> > > | **Task** | BC| TD3+BC | CQL | IQL |DOGE| POR  | TSRL| TELS (using the same set of hyper-parameters) | TELS (reported in the paper)
> > > | ---------- |----------- | ----------- | ----------- | ----------- | ----------- | ----------- | ----------- | ----------- | ----------- |
> > > | Hopper-m   | 29.9 $\pm$  11.7 | 40.1 $\pm$ 18.6 | 43.1 $\pm$  24.6| 46.7 $\pm$ 6.5 | 44.2 $\pm$ 10.2    |46.4 $\pm$ 1.7 |62.0 $\pm$ 3.7  | 65.5 $\pm$ 12.4 | 77.3 $\pm$ 10.7
> > > | Hopper-m-r   | 12.1 $\pm$ 5.3 | 7.3 $\pm$ 6.1 | 2.3 $\pm$ 1.9 | 13.4 $\pm$ 3.1 | 17.9 $\pm$ 4.5 | 17.4 $\pm$ 6.2   | 21.8 $\pm$ 8.2 | 30.1 $\pm$ 7.6 | 43.2 $\pm$ 3.5
> > > | Hopper-m-e  |27.8 $\pm$ 10.7 | 17.8 $\pm$ 7.9 | 29.9 $\pm$ 4.5 | 34.3 $\pm$ 8.7 | 50.5 $\pm$ 25.2 | 37.9 $\pm$ 6.1  |50.9 $\pm$ 8.6  | 71.5 $\pm$ 9.1 | 100.9 $\pm$ 6.8
> > >
> > > - As the reviewer can observe from the above results, even using a single set of hyper-parameters, TELS still achieves much better performance as compared to all other offline RL baseline methods.
> > > - In our original submission, we used the same set of hyper-parameters for training all Halfcheetah and Walker2d tasks, as both of them share identical state and action dimensions (17 for states and 6 for actions). In contrast, Hopper tasks have a smaller state-action space (11 states and 3 actions), making it comparatively simpler given the same amount of training data (e.g., 10k samples). Consequently, we opted for a more aggressive learning strategy for Hopper tasks, i.e., using higher $\alpha=0.1$ to focus more on value maximization. In practice, for tasks with relatively abundant offline data and higher state-action space coverage, we suggest the practitioner can reasonably relax the regularization strength (e.g., increase $\alpha$) to enable even better performance of TELS.

---

### Official Review · Reviewer_G8R6 · 2024-10-31

**Soundness:** 2
**Presentation:** 2
**Contribution:** 2
**Rating:** 3
**Confidence:** 3

**Summary:**

This paper proposes (1) a T-symmetry enforced dynamics model, and a (2) a advantage weighted next latent state predictor + inverse dynamics models to create an offline RL algorithms. The goal is to use both of these inductive biases to improve the sample efficiency in offline RL.

**Strengths:**

The quality of writing and clarity is good throughout. The main strength of this paper seem to be the results in Figure 1 on Ant-maze datasets, but I think this is partly because the baselines are chosen are just not good enough (more in questions and weaknesses).

**Weaknesses:**

1) The primary weakness is the complexity and the number of moving parts of this paper. The algorithm contains:
    (a)  state encoder
    (b)  state decoder
    (c)  action encoder
    (d)  action decoder
    (e) latent inverse dynamics model
    (f)  latent ODE forward model
    (g) latent ODE reverse model
    (h) latent state value function
    (i)  latent stochastic guide-policy
  As a result of this, the number of loss function terms is also very high (>6 as far as I can tell), making it extremely unclear why the algorithms is actually working. Although complex problems can justify complex architectures and loss functions, I do not think antmaze / other toy offline RL settings warrant such a arduous setup.

2) In Table 1, I think there are many important baselines missing - Goal condition Behavior cloning, HIQL (which is cited multiple times in the paper) or other hierarchical / model based methods like the decision diffuser.

**Questions:**

On line 148, you mention -- "making Q-function maximization the only option for policy optimization, which inevitably requires adding conservative action-level constraints", but your algorithm uses the same conservatism while training the guiding policy using advantage weighted behavior cloning. Could you clarify this?

---

> ### Author Response · Authors · 2024-11-23
> **Response to Reviewer G8R6 (Part 1/2)**
>
> We thank the reviewer for the thoughtful review and comments. Regarding the concerns from the reviewer, we provide the detailed responses as follows:
>
> > **W1. The primary weakness is the complexity and the number of moving parts of this paper ... I do not think antmaze / other toy offline RL settings warrant such a arduous setup.**
>
> - We thank the reviewer for this thoughtful comment. Although our framework looks complex, however, its design philosophy is clear and straightforward: we first extract generalizable state representations from the T-symmetry enforced inverse dynamics model (TS-IDM), and then conduct offline policy optimization within the latent space to enable state-stitching and maximum degree of small-sample generation.
> - Specifically, although there are many loss terms in the training objective in our TS-IDM, we find that it is easy to learn and enjoys stable convergence. If the reviewer checks the detailed design and each loss term in TS-IDM, you can find that every component is closely coupled during learning. For example,
>     - Both the latent ODE forward and reverse dynamics models ($h_{fwd}$, $h_{rvs}$) need the latent actions $z_a$ from the latent inverse dynamics model $h_{inv}$, hence their loss terms $\ell_{fwd}$, $\ell_{rvs}$, and $\ell_{inv}$ in Eq. (5), (6) and (4) are strongly coupled.
>     - The state encoder $\phi_s(s)$ and decoder $\psi_s(z_s)$ not only need to achieve reasonable state reconstruction ($\ell_{s-rec}$ in Eq. (3)), but also need to satisfy the ODE property as required in the latent ODE forward and reverse dynamics models ($h_{fwd}$, $h_{rvs}$), which is enforced in their loss terms $\ell_{fwd}$, $\ell_{rvs}$ and $\ell_{s-ode}$. Hence the loss term $\ell_{s-rec}$ in Eq. (3) is also strongly coupled with loss terms in Eq. (5), (6), (7).
>     - Lastly, we also enforce the T-symmetry consistency ($\ell_{T-sym}$ in Eq. (8)) between the latent ODE forward and reverse dynamics models ($h_{fwd}$, $h_{rvs}$), which makes the loss terms Eq. (5), (6), (8) also strongly coupled.
> - As the reviewer can observe from the above, the design of TS-IDM ensures all components and loss terms are strongly coupled with each other, which allows them can be consistently learned without suffering from conflicts as observed in most multi-task learning problems. In fact, we find TS-IDM is very easy to train in our experiments. In all our MuJoCo locomotion tasks and antmaze tasks, we only use a single set of hyperparameters to train TS-IDM. We also provide all the learning curves of TS-IDM and our offline policy optimization procedure in Appendix C, in which the reviewer can clearly see its stable learning curves.
> - Moreover, we also find that our proposed TS-IDM is even easier to train than the T-symmetry enforced dynamics model (TDM) proposed in TSRL [1]. In the TSRL paper, their authors mentioned that the TDM needs to first pre-train the encoder and decoders before joint training on other elements, otherwise will suffer from learning stability issues. However, in our proposed TS-IDM, we do not need to pre-train the encoder and decoders. All its components can be jointly learned in a single stage, due to its strong internal strong consistency among all of its components.
> - Lastly, we want to highlight that all our experiments are conducted on very small offline datasets (as few as 1% of the original D4RL tasks), which are not toy tasks but exceptionally challenging for offline policy learning. Most existing offline RL algorithms fail miserably even on MuJoCo and antmaze tasks when the data size is significantly reduced (see results in our paper as well as [1], [2]). By contrast, our proposed TELS significantly outperforms all other baselines, even the existing SOTA small-sample offline RL method TSRL by a large margin. Notably, TSRL completely fails in 0.1M Antmaze medium and large tasks, but our method still achieves strong performance even with limited training samples.

---

> ### Author Response · Authors · 2024-11-23
> **Response to Reviewer G8R6 (Part 2/2)**
>
> > **W2. In Table 1, I think there are many important baselines missing - Goal condition Behavior cloning, HIQL or other hierarchical / model based methods like the decision diffuser.**
>
> - HIQL is actually a hierarchical, goal-conditioned extension of POR [3] which shares almost the same algorithm architecture, but changes the guide-policy to predict longer horizon sub-goals instead of predicting the next optimal state as in POR, as well as migrating to the goal-conditioned RL setting. This is also admitted in the HIQL paper, as their authors mentioned in their related work, "our method is closely related to POR". As our problem setting is the standard offline RL rather than goal-conditioned RL, and many of our experiments are conducted on tasks that do not have a specific goal target (e.g., MuJoCo tasks), hence we compare with the more relevant POR rather than HIQL.
> - Second, we'd like to clarify that our approach is very different from the model-based RL methods. Although we also learned a T-symmetry enforced inverse dynamics model (TS-IDM), this model is **never used to generate model rollout data**. TS-IDM is only used to learn generalizable state representations and provide the T-symmetry consistency measures for downstream offline policy learning. This is completely different from the model-based RL, in which the models are used to generate synthetic rollout data during policy learning. Moreover, as shown in the TSRL paper [1], model-based offline RL methods like MOPO [4] perform quite poor in the small dataset setting, as the limited sample cannot support learning a dynamics model with reasonable accuracy, and the high approximation error in the model rollout data severely deteriorate offline policy learning performance. Therefore, we focus more on comparing model-free offline RL baselines, which generally perform better in the small sample setting.
> - Lastly, we did not compare offline RL methods with heavy model architecture (e.g., transformer or diffusion-based methods) in our original submission, as we found these methods are simply not able to be learned well under small dataset setting (i.e., tasks with only 10k data samples) and do not produce any meaningful policy in our experiments. These methods are not sample-efficient and need a large amount of data to train in order to achieve reasonable performance.
>
> > **Q1. On line 148, you mention -- "making Q-function maximization the only option for policy optimization, which inevitably requires adding conservative action-level constraints", but your algorithm uses the same conservatism while training the guiding policy using advantage weighted behavior cloning. Could you clarify this?**
>
> Note that the policy learning scheme in Eq. (12) is an "AWR-style" objective rather than a typical "advantage weighted behavior cloning" mentioned by the reviewer. If the reviewer checks our policy learning objectives in Eq. (11) and (12), they are formulated completely within the latent state space, in other words, we only restrict the deviation of the next state output of guide-policy to the latent state in the data. There is completely no action involved in this case, thus completely removing the need to add any potentially conservative action-level regularizations.
>
>
> **References**
>
> [1] Cheng, P., et al. Look beneath the surface: Exploiting fundamental symmetry for sample-efficient offline RL. NeurIPS 2023.
>
> [2] Li, J., et al. When Data Geometry Meets Deep Function: Generalizing Offline Reinforcement Learning. ICLR 2023.
>
> [3] Xu, H., et al. A policy-guided imitation approach for offline reinforcement learning. NeurIPS 2022.
>
> [4] Yu, T., et al. MOPO: Model-based Offline Policy Optimization. NeurIPS 2020.

---

> ### Author Response · Authors · 2024-11-26
> **A gentle reminder for reviewer**
>
> Dear reviewer G8R6,
>
> As the rebuttal phase is coming to a close, we wanted to check back to see whether you have any feedback or remaining questions. We would be happy to clarify further, and grateful for any other feedback you may provide.
>
> We really appreciate your time engaged in the review process and look forward to your replies!
>
> Best regards,
>
> Authors of Paper 10618

---

### Author Response · Authors · 2024-11-23
**General Response and Revision Summary**

We thank all the reviewers for their detailed and constructive comments. We have conducted additional experiments and revised our paper (highlighted in blue text color) to address the concerns of the reviewers. The modifications are summarized as follows:

- (For reviewer g1ie and 3YZb) We have added the accidentally omitted loss term $\ell_{inv}$ in the complete training loss of TS-IDM (Eq. (9)) in our revised paper.
- (For reviewer g1ie and dspK) We provided a series of additional ablation results in Appendix B.2 of our revised paper, including ablations for hyperparameter $\beta$ and $\eta$, quality of learned representations, and impact of T-symmetry consistency regularization on guide-policy optimization.
- (For reviewer g1ie) We added additional results on 0.1M Antmaze-medium-diverse tasks with different deletion ratios. We also added comparative results with baseline methods IQL, DOGE, and POR. The new results are updated in Appendix B.3.
- (For reviewer G8R6 and dspK) We added all learning curves of our proposed TS-IDM and the offline policy learning procedure in TELS in Appendix C, to demonstrate the learning stability of our method.

---

### Meta-Review · Area_Chair_SjEg · 2024-12-22

**Metareview:**

This paper introduces a T-symmetry enforced inverse dynamics model (TS-IDM) into offline reinforcement learning. Overall the proposed TS-IDM is a latent space inverse dynamics model, which exploits the time-reversal symmetries of the underlying dynamics system to regularize the derived representations.

I agree with reviewer 3YZb in regards to limited novelty of this approach and with Reviewer G8R6 in regards to complexity of the approach (and a lack of very rigorous ablations: not just on the hyperparameters, but also on the architectures and such). I think the bar for a somewhat more complex approach is higher.

Unfortunately, taking into account the points raised by the reviewers and the discussion, we cannot accept this paper for now.

**Additional Comments On Reviewer Discussion:**

The points raised by the reviewers are regarding hyperparameters, novelty and significance of contribution, and the complexity of the system. I generally agree with these points, and while the results of the paper in the small dataset regime are strong, I believe more work is needed to make sure that this method is adopted and is of significance to the community.

---

### Decision · Program_Chairs · 2025-01-22

Reject